# X-Mahalanobis: Transformer Feature Mixing for Reliable OOD Detection

**Tong Wei**[1,2,3]   **Bo-Lin Wang**[1,2]   **Jiang-Xin Shi**[3,4]
**Yu-Feng Li**[3,4]   **Min-Ling Zhang**[1,2]

[1]School of Computer Science and Engineering, Southeast University, Nanjing, China
[2]Key Laboratory of Computer Network and Information Integration (Southeast University),
Ministry of Education, China
[3]National Key Laboratory for Novel Software Technology, Nanjing University, China
[4]School of Artificial Intelligence, Nanjing University, China
`{weit,wangbl}@seu.edu.cn`

## Abstract

Recognizing out-of-distribution (OOD) samples is essential for deploying robust machine learning systems in open-world environments. While conventional OOD detection approaches rely on feature representations from the penultimate layer of neural networks, they often overlook informative signals embedded in intermediate layers. In this paper, we present a straightforward feature mixing approach for pretrained Transformers, which combines multi-layer representations via calculated importance weights, and identifies OOD samples using Mahalanobis distance in the blended feature space. When in-distribution samples are accessible, we show that parameter-efficient fine-tuning strategies effectively balance classification accuracy and OOD detection performance. We conduct extensive empirical analyses to validate the superiority of our proposed method under zero-shot, and fine-tuning settings using both class-balanced and long-tailed datasets. The source code is available at `https://github.com/SEUML/X-Maha`.

## 1   Introduction

In recent years, deep learning models have made significant progress in various domains [40, 24]. However, a critical issue with these models is their tendency to be overly confident in their predictions, even when the input deviates greatly from the data distribution seen during training. This issue underscores the need for effective out-of-distribution (OOD) detection when training deep neural networks (DNNs). The detection of OODs is crucial to ensure the safety of the model in many applications, such as medical diagnostics [43], industrial inspection [3], and autonomous driving [26]. For example, in the field of medical imaging, DNNs may fail to provide an accurate diagnosis when presented with data that falls outside the training data distribution, such as images from an unknown scanner. Therefore, it is imperative for a reliable model not only to recognize in-distribution (ID) samples, but also to flag any OOD input as "unknown".

Existing OOD detection methods design various scoring functions to assign an input sample a likelihood to be OOD, using 1) *predicted probabilities* [14, 33, 31, 9, 32], 2) *output logits* [48, 2], and 3) *learned features* [25, 9, 36] by the model. However, these approaches neglect the rich information in the features learned by the layers of shallow neural networks. Our motivation stems from the observation that while the final features of a neural network are nonlinear transformations of shallow features and inherently retain some information from earlier layers, features extracted from different layers provide diverse representations of the data. Given that certain features may be particularly effective for distinguishing between ID and OOD samples, it is crucial to comprehensively leverage

39th Conference on Neural Information Processing Systems (NeurIPS 2025).

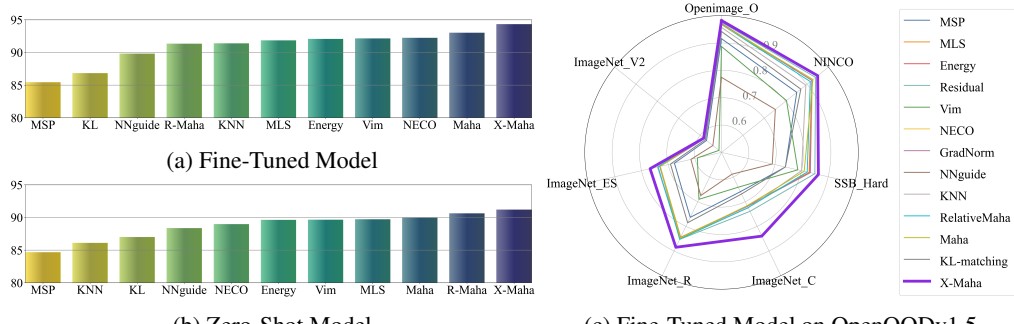

(a) Fine-Tuned Model

(b) Zero-Shot Model

(c) Fine-Tuned Model on OpenOODv1.5

Figure 1: (a-b): AUROC of X-Maha and competing methods based on fine-tuned/zero-shot models. The experimental settings are the same as in Table 2. We denote RelativeMaha as R-Maha. (c) AUROC of X-Maha and competing methods on OpenOODv1.5 benchmark using fine-tuned model.

the information from all layers to enhance OOD detection performance. While the motivation is appealing, a core challenge remains: *how to effectively utilize shallow layer features for OOD detection?*

To address the above issue, we propose a new OOD detection approach by leveraging features from all layers with an adaptive fusion module. We draw inspiration from the geometric properties of "neural collapse" [37], which states that the convergence of within-class covariance approaches zero in the terminal phase of training as each activation collapses toward its respective class mean. Therefore, we propose to measure the total variance of features across different layers of the neural network to describe their importance weights for OOD detection. Layers with larger total variance have more influence, while the contribution of layers with smaller total variance is down-weighted. The advantage of this method is that layer weights are computed based on the data, without the need for manual parameter tuning. Using the weighted fused features, we calculate the Mahalanobis distance between the test sample and the data distribution of each ID class to calculate its OOD score.

Furthermore, we fine-tune the pre-trained visual models, including Vision Transformer (ViT) [8] and CLIP [39], using in-distribution data to adapt the feature representations to down-stream tasks. We empirically find that parameter-efficient fine-tuning strategies consistently outperforms full parameter fine-tuning and are more robust to hyperparameter choice, which coincides with prior works [44, 11]. Specifically, by freezing the pre-trained model and adding a small number of learnable parameters. Based on this finding, we develop a general fine-tuning framework and implement all comparison methods within this framework in our experiments. We also conducted an in-depth analysis of various fine-tuning strategies. Figure 1 presents the results for in-distribution samples (from ImageNet) processed by ViT-B/16 under various experimental settings. Our X-Maha (X-Mahalanobis) consistently achieves state-of-the-art OOD detection performance in both fine-tuned and zero-shot scenarios, and demonstrates superior performance on the challenging OpenOODv1.5 benchmark.

To systematically evaluate our approach, we focus on both class-balanced ID datasets, which are commonly used in existing OOD detection literature [31, 48, 2], and long-tailed ID datasets because the distribution of real-world data is often imbalanced and highly skewed on a per-class basis, with a majority of classes containing a small number of samples [53, 52, 61]. Notably, long-tailed OOD detection has been studied in several recent works by improving 1) *representation learning* [49, 54, 51, 5], and 2) *probabilistic calibration* [22, 35]. However, these methods often require the use of OOD data to train the model. In contrast, our approach only requires fine-tuning the model using ID data, and more importantly, with no changes needed for the proposed feature mixing module.

Our contributions are summarized as follows:

1. We propose a new OOD detection method that exploits features from shallow layers of pre-trained Transformers to enhance OOD separation.

2. We propose a simple but effective strategy to fuse multiple layer features with the importance weights by measuring the covariance of features in each layer.

3. We justify the effectiveness of the proposed method in zero-shot setting, and fine-tuning settings using both class-balanced and long-tailed datasets. Additionally, we show that the propose method can generalize to various fine-tuning strategies and pre-trained models.

## 2 Related Works

**Out-of-distribution detection.** In recent years, the field of OOD detection has gained considerable attention. The Maximum Softmax Probability (MSP) method [13] serves as a foundational baseline, utilizing softmax predictions as OOD scores. Building on this, ODIN [30] improves the softmax score by perturbing input data and rescaling logits, enhancing its effectiveness in distinguishing OOD samples. Further advancements explore alternative scoring mechanisms, such as the energy score [31], which is further refined through feature clipping in ReAct [45]. Additionally, gradient-based approaches have been explored to differentiate between ID and OOD data [19, 1]. Among previous studies, the use of the Mahalanobis distance has shown significant promise. A prior work [28] proposes to ensemble the Mahalanobis distance score calculated by features of each layer and determine the optimal ensemble weights using an auxiliary OOD validation dataset. Trusted [7] introduces a novel approach that combines feature fusion during training with the Mahalanobis distance during testing, guided by the optimal transport principle. On top of the CLIP model, CLIPN [50] learns a "no" prompt to capture the negation-semantic with images using an auxiliary dataset, and performs OOD detection depending on the similarity between the input image and the "no" prompt. Similarly, NegLabel [23] extracts potential negative labels from a corpus database and employs zero-shot CLIP for OOD detection by combining ID classes and negative labels.

**Long-tailed out-of-distribution detection.** In long-tailed OOD detection, prior research has examined several strategies to mitigate the challenges posed by class imbalance, including the use of oversampling techniques and threshold adjustments to improve performance [29]. Open Sampling [51] incorporates OOD data to address the class imbalance problem. PASCL [49] focuses on enhancing representation learning for tail classes by leveraging a contrastive learning method, helping to improve the separation between minority classes and OODs. Prior work [22] identifies several common scenarios where the OOD-to-ID probabilities should be the ID-class-prior distribution and proposes two strategies to modify existing inference-time detection methods. EAT [54] proposes expanding the class space of ID classes with virtual classes to tackle OOD data. COCL [35] introduces a calibrated learning approach aimed at improving outlier class detection in long-tailed tasks.

**Parameter-efficient fine-tuning.** PEFT methods freeze the pre-trained model and introduce only a few learnable parameters for adaptation, which can effectively reduce overfitting and accelerate convergence. Adapter [8] introduces a bottleneck module to optimize only a small subset of parameters. BitFit [58] focuses on fine-tuning only the bias terms of the model, significantly reducing the number of parameters that need to be updated during training. VPT [21] prepends learnable prompts at each layer, offering two versions: VPT-Shallow, which uses prompts at shallow layers, and VPT-Deep, which applies them across deeper layers. LoRA [17] further optimizes efficiency by applying low-rank adaptations, minimizing the overall parameter count while retaining performance. AdaptFormer [4] builds on the Adapter method by shifting from a sequential to a parallel design. LIFT [44] provides an empirical analysis showing that the commonly used full fine-tuning strategy is prone to overfitting, especially on long-tailed datasets.

## 3 Method

In this section, we present a simple Mahalanobis-based OOD detection method by mixing features from all Transformer layers based on importance weight.

### 3.1 Preliminary

We first introduce the problem setting and notations used throughout this paper.

1. We denote the training set as $\mathcal{D}_{train} = \{(\boldsymbol{x}_i, y_i)\}_{i=1}^{N}$, where $\boldsymbol{x}_i \in \mathbb{R}^d$ represents an input image, $y_i \in [C]$ denotes its ground-truth class label, and $C$ denotes the total number of classes in the training set. At test time, our goal is to flag images that do not belong to any of the training classes using our OOD detector.

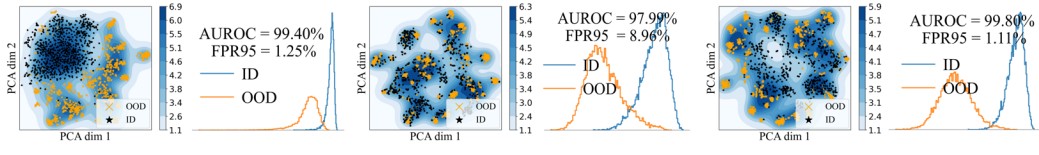

| (a) Maha on 11-th Layer | (b) Maha on Penultimate Layer | (c) X-Maha |

Figure 2: X-Maha: We illustrate how to improve Mahalanobis-based OOD detection. (a) Mahalanobis distance applied to the 11-th layer. (b) Mahalanobis distance applied to the penultimate layer features. (c) X-Maha which is applied to all layer features. Each subfigure comprises two components: a feature visualization map and the corresponding OOD score distribution of test data. The visualizations are based on data sampled from CIFAR-100 (ID) and Tiny ImageNet (OOD) using fine-tuned ViT-B/16.

2. Without loss of generality, let the deep neuron network be $F = f \circ g$, where $f(\cdot)$ is known as the feature exactor and $g(\cdot)$ is the classifier. For each layer $\phi(\cdot)$ in $f$, we define the transformation learned by the $l$-th layer as $\phi_l(\cdot)$. For an instance $\boldsymbol{x}$, its output from the $l$-th layer is denoted as $\boldsymbol{x}^l = \phi_l(\boldsymbol{x})$. In particular, we denote the final feature learned by the model $\boldsymbol{x}^L = \phi_L(\boldsymbol{x})$, where $L$ denotes the number of Transformer layers.

3. In this paper, we build our OOD detector based on the Mahalanobis distance. For any test image $\boldsymbol{x}$, we calculate the negative distance between the image feature $f(\boldsymbol{x})$ and feature distribution of each class as the scoring function:

$$M(\boldsymbol{x}; \boldsymbol{\mu}_c, \boldsymbol{\Sigma}) = -\left(f(\boldsymbol{x}) - \boldsymbol{\mu}_c\right)^\top \boldsymbol{\Sigma}^{-1} \left(f(\boldsymbol{x}) - \boldsymbol{\mu}_c\right), \tag{1}$$

where $\boldsymbol{\mu}_c$ is the mean feature vector of class $c$ and $\boldsymbol{\Sigma}$ is the covariance matrix of ID data.

4. To measure the Mahalanobis distance, we calculate the empirical class mean and covariance matrix of training samples as follows:

$$\boldsymbol{\mu}_c = \frac{1}{N_c} \sum_{i:y_i=c} f(\boldsymbol{x}_i), \boldsymbol{\Sigma} = \frac{1}{N} \sum_{c=1}^{C} \sum_{i:y_i=c} \left(f(\boldsymbol{x}_i) - \boldsymbol{\mu}_c\right)\left(f(\boldsymbol{x}_i) - \boldsymbol{\mu}_c\right)^\top, \tag{2}$$

where $N_c$ is the number of training samples with class $c$. This is equivalent to fitting the class-conditional Gaussian distribution with a tied covariance to the training samples under the maximum likelihood estimator [28].

## 3.2 X-Maha: Feature Mixing for Mahalanobis-based OOD Detection

By default, the Mahalanobis distance in Eq. (1) uses the final output of the feature extractor, i.e., $f(\boldsymbol{x})$, neglecting rich information in shallow layer features. Therefore, we now proceed to present our approach to demonstrate that shallow features can help improve OOD detection performance. For any test image $\boldsymbol{x}$ and a fine-tuned model, we first obtain its hidden representations $\boldsymbol{x}_i^l$ of the $l$-th layer, $\forall 1 \le l \le L$. Notably, we may use "features" and "representations" interchangeably throughout the paper. We then integrate features from all layers by different importance weights. Formally, we compute the fused feature representation of $\boldsymbol{x}$ by:

$$\Phi(\boldsymbol{x}) = \sum_{l=1}^{L} \alpha^l \boldsymbol{x}^l, \tag{3}$$

where $\alpha^l$ is the weight of the $l$-th layer. To measure the Mahalanobis distance, we also calculate the class mean feature vectors and global covariance matrix in the fused feature space. We reformulate Eq. (2) by fusing shallow features as follows:

$$M_{\text{X-Maha}}(\boldsymbol{x}; \widetilde{\boldsymbol{\mu}}_c, \widetilde{\boldsymbol{\Sigma}}) = -\left(\Phi(\boldsymbol{x}) - \widetilde{\boldsymbol{\mu}}_c\right)^\top \widetilde{\boldsymbol{\Sigma}}^{-1} \left(\Phi(\boldsymbol{x}) - \widetilde{\boldsymbol{\mu}}_c\right), \tag{4}$$

where $\widetilde{\boldsymbol{\Sigma}} = \frac{1}{N} \sum_{c=1}^{C} \sum_{i:y_i=c} \left(\Phi(\boldsymbol{x}_i) - \widetilde{\boldsymbol{\mu}}_c\right)\left(\Phi(\boldsymbol{x}_i) - \widetilde{\boldsymbol{\mu}}_c\right)^\top$ and $\widetilde{\boldsymbol{\mu}}_c = \frac{1}{N_c} \sum_{i:y_i=c} \Phi(\boldsymbol{x}_i)$.

Figure 2 provides an intuitive example in which shallow features can exhibit better discriminativity between ID and OOD data than the final layer features. By mixing features as in Eq. (3), X-Maha can effectively alleviate feature overlapping between ID and OOD data.

We now provide a simple way to set $\alpha^l$. To reflect the importance of each layer, we propose to calculate the weights by measuring the instance-discrimination capacity or variability of the features.

**Definition 3.1** (Measure of Variability). Given a collection of $\boldsymbol{x}_l$, we calculate the mean feature by $\boldsymbol{\mu}^l = \frac{1}{N} \sum_{i=1}^{N} \boldsymbol{x}_i^l$, then and measure the feature variability of the $l$-th layer by:

$$\alpha^l = \mathrm{Tr}((\boldsymbol{A}^l)^\top \boldsymbol{A}^l), \tag{5}$$

where $\boldsymbol{A}^l = (\boldsymbol{x}_1^l - \boldsymbol{\mu}^l, \boldsymbol{x}_2^l - \boldsymbol{\mu}^l, \cdots, \boldsymbol{x}_N^l - \boldsymbol{\mu}^l)^\top$ is the centralized feature matrix of the $l$-th layer, and $\mathrm{Tr}(\cdot)$ denotes the trace of a matrix, which is the sum of its diagonal elements. We normalize the weights so that the sum of the weights across all layers is equal to 1.

The trace of the matrix $\mathrm{Tr}((\boldsymbol{A}^l)^\top \boldsymbol{A}^l)$ is proportional to the total variance of the features in the $l$-th layer. A higher value of this trace indicates that the features at this layer are, on average, more spread out across the training samples. This substantial variability suggests that the layer captures diverse and discriminative patterns, making it highly sensitive to differences between instances. Therefore, assigning a higher weight to such layers during feature fusion amplifies the contribution of these more informative representations. Notably, Eq. (5) presents one simple way to set mixing weights, though not necessarily optimal. We leave further optimization for future work, as our focus here is on demonstrating the effectiveness of mixing features from shallow layers.

**Distinctions with prior works.** Our work differs from *Mahalanobis* [28] and *Trusted* [7], which also use internal representations. 1) *Mahalanobis* calculates the OOD score using the representation of each layer individually and weights them together by training a logistic regression model using the validation set. Our approach computes importance weights from training data and does not require any validation set. 2) *Trusted* treats every layer equally with the same importance and averages the representations. It is clear that certain layer representations may be more effective in detecting OODs, whereas others may bring noise. Our approach can prevent the degradation of the overall OOD detection performance even in the case when the features from some layers are not effective: the weights would be nearly zero for those ineffective layers.

## 3.3 On the Fine-tuning Strategy for OOD Detection

**Parameter-efficient fine-tuning is more robust than fully fine-tuning.** To adapt the pre-trained models to downstream classification and OOD detection tasks, we learn a linear classifier and fine-tune the feature extractor using ID training data. In this paper, we adopt the logit adjustment loss [34] as the optimization objective for its simplicity and good generalization ability. The key advantage of this choice is that, for class-balanced ID datasets, it simplifies to the conventional cross-entropy loss; however, for long-tailed ID datasets, it allows the model to balance predictive confidence across classes. Formally, the logit adjustment loss is defined as:

$$\mathcal{L}_{\mathrm{LA}}(\boldsymbol{x}, y = j) = -\log \frac{\exp(z_j + \log \mathrm{P}(y = j))}{\sum_{k \in [C]} \exp(z_k + \log \mathrm{P}(y = k))} \tag{6}$$

where $y = j$ denotes the ground-truth label of the input $\boldsymbol{x}$, and $z_j$ is the logit (pre-softmax activation) for class $j$. The class-prior probability $\mathrm{P}(y = j)$ is estimated from the training distribution.

However, when choosing the fine-tuning strategy, we observe that full parameter fine-tuning (FFT) is significantly more sensitive to hyperparameters, such as learning rate, compared to parameter-efficient fine-tuning (PEFT), especially when the ID data follows a long-tailed label distribution. Figure 3 highlights the impact of learning rates on both fine-tuning strategies in CIFAR-100 (ID) classification accuracy and OOD detection AUROC, averaged on six OOD datasets. The $x$-axis denotes the learning rate.

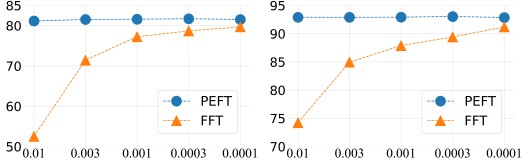

(a) ImageNet-LT ACC    (b) ImageNet-LT AUROC

Figure 3: Comparison of the sensitivity of FFT and PEFT to learning rate.

The results indicate that FFT requires careful tuning of learning rates to achieve optimal performance, while PEFT demonstrates more robust performance across a wider range of hyperparameters. Moreover, FFT necessitates tuning hyperparameters like the learning rate individually for each dataset, whereas PEFT allows for consistent hyperparameter settings across multiple datasets, reducing the burden of hyperparameter search.

**Extension of our approach to vision-language models.** Notably, our proposed X-Maha approach is model-agnostic and can be used for CLIP-like models. Specifically, we calculate the cosine similarity between the image embedding and ID class text prompt embeddings with minimal computational overhead. This similarity score is integrated into X-Maha to improve the effectiveness of OOD detection. Formally, the revised scoring function is defined as follows: $G(\boldsymbol{x}) = \max_{c \in [C]} M_{\text{X-Maha}}(\boldsymbol{x}; \widetilde{\boldsymbol{\mu}}_c, \widetilde{\boldsymbol{\Sigma}}) + \lambda \cdot \text{sim}(\boldsymbol{v}, \boldsymbol{t}_c)$, where $\boldsymbol{v}$ denotes the image embedding of $\boldsymbol{x}$ extracted by the pre-trained image encoder, and $\boldsymbol{t}_c$ represents the text prompt embedding of class $c$, i.e., both image and text embeddings are obtained from pre-trained CLIP. The similarity measure $\text{sim}(\boldsymbol{v}, \boldsymbol{t}_c)$ is defined as: $\text{sim}(\boldsymbol{v}, \boldsymbol{t}_c) = \frac{e^{\boldsymbol{v}^\top \boldsymbol{t}_c}}{\sum_k e^{\boldsymbol{v}^\top \boldsymbol{t}_k}}$, where we use the default prompt template "a photo of a {classname}" to obtain text embedding $\boldsymbol{t}_c$ in our experiments. The hyperparameter $\lambda$ controls the relative influence of the predicted similarity scores of the vision-language model. *Notably, we set $\lambda = 0$ when using vision-only models.* A test image is classified as OOD if $G(\boldsymbol{x}) \geq \rho$, where $\rho$ is selected such that a high proportion of ID data exceeds this threshold. For samples classified as ID, the class label is determined as $\hat{y} = \arg\max_{c \in [C]} p_c$, where $\boldsymbol{p} = F(\boldsymbol{x})$ denotes the predicted class probabilities from the classifier.

## 4 Experiments

We extensively evaluate X-Maha across different datasets and pre-trained models. Due to space constraints, in the main paper, we report the experimental results of models fine-tuned on class-balanced or long-tailed ID datasets.

### 4.1 Experiments Setup

In this section, we compare our approach with the latest algorithms across both small- and large-scale OOD detection benchmarks. In line with prior research, we utilize CIFAR-100 and ImageNet as the in-distribution (ID) datasets. Additionally, we incorporate the more challenging long-tailed variants, CIFAR-100-LT and ImageNet-LT, as ID training sets to further demonstrate the effectiveness of our proposed method in OOD detection scenarios in the appendix. The imbalance ratio for CIFAR-100-LT is set to 100, reflecting a highly imbalanced class distribution.

**OOD datasets.** When CIFAR-100 or CIFAR-100-LT is used as the ID dataset, we evaluate OOD detection performance on a range of diverse datasets, including Textures [6], SVHN [57], CIFAR-10,

Table 1: OOD detection performance on CIFAR-100 (ID) and six OOD datasets.

| Method | Texture | | SVHN | | CIFAR10 | | Tiny ImageNet | | LSUN | | Places365 | | Average | |
|---|---|---|---|---|---|---|---|---|---|---|---|---|---|---|
| | AUROC | FPR95 | AUROC | FPR95 | AUROC | FPR95 | AUROC | FPR95 | AUROC | FPR95 | AUROC | FPR95 | AUROC | FPR95 |
| IMAGENET-21K PRE-TRAINED VIT | | | | | | | | | | | | | | |
| MSP | 97.65 | 11.81 | 94.91 | 28.17 | 94.92 | 26.32 | 88.58 | 44.50 | 86.75 | 64.21 | 92.23 | 41.41 | 92.51 | 36.07 |
| MLS | 99.79 | 0.83 | 97.38 | 10.31 | 97.07 | 13.42 | 93.28 | 25.16 | 98.09 | 10.93 | 98.98 | 5.39 | 97.43 | 11.01 |
| Energy | 99.86 | 0.57 | 97.48 | 9.47 | 97.09 | 12.88 | 93.51 | 23.61 | 98.59 | 7.58 | 99.26 | 3.78 | 97.63 | 9.65 |
| Mahalanobis | 99.97 | 0.12 | 99.16 | 3.92 | 97.09 | 16.49 | 97.99 | 8.96 | 99.61 | 1.07 | 99.67 | 1.33 | 98.92 | 5.32 |
| Residual | 99.99 | 0.02 | 97.66 | 12.81 | 92.08 | 41.38 | 99.10 | 3.68 | 99.93 | 0.00 | 99.92 | 0.08 | 98.12 | 9.66 |
| Vim | 99.89 | 0.44 | 97.68 | 8.63 | 97.13 | 12.73 | 94.09 | 21.96 | 99.39 | 5.72 | 99.39 | 2.94 | 97.84 | 8.74 |
| NECO | 99.83 | 0.83 | 97.95 | 8.70 | 97.31 | 13.98 | 94.25 | 21.93 | 98.29 | 10.77 | 99.08 | 5.35 | 97.78 | 10.26 |
| Trusted | 100.0 | 0.00 | 98.78 | 5.77 | 93.35 | 33.51 | 98.09 | 9.76 | 100.0 | 0.01 | 100.0 | 0.00 | 98.37 | 8.17 |
| KL-matching | 98.60 | 6.10 | 96.66 | 14.93 | 96.34 | 17.12 | 90.05 | 34.17 | 88.15 | 49.34 | 93.67 | 28.21 | 93.91 | 24.98 |
| NNguide | 99.24 | 3.03 | 98.70 | 5.00 | 97.12 | 17.42 | 92.48 | 28.02 | 93.44 | 40.52 | 96.25 | 21.11 | 96.21 | 19.18 |
| RelativeMaha | 98.25 | 6.35 | 97.44 | 12.72 | 96.41 | 17.73 | 91.66 | 34.44 | 90.66 | 49.66 | 94.49 | 30.37 | 94.82 | 25.21 |
| KNN | 99.19 | 3.10 | 98.51 | 6.13 | 96.42 | 20.66 | 91.56 | 30.20 | 92.79 | 45.19 | 96.06 | 21.73 | 95.76 | 21.17 |
| X-Maha (ours) | 100.0 | 0.00 | 99.50 | 1.91 | 96.47 | 19.52 | 99.80 | 1.11 | 100.0 | 0.00 | 100.0 | 0.00 | 99.29 | 3.76 |
| CLIP-VIT-B/16 | | | | | | | | | | | | | | |
| MSP | 91.14 | 41.33 | 86.22 | 57.75 | 87.35 | 53.18 | 82.11 | 62.50 | 74.83 | 80.64 | 84.02 | 60.61 | 84.28 | 59.33 |
| MLS | 96.11 | 20.73 | 91.58 | 41.81 | 93.32 | 30.69 | 88.58 | 45.86 | 88.49 | 51.20 | 93.15 | 33.12 | 91.87 | 37.23 |
| Energy | 96.56 | 18.03 | 91.85 | 41.92 | 93.77 | 28.89 | 89.06 | 44.49 | 89.66 | 45.66 | 93.92 | 29.21 | 92.47 | 34.70 |
| Mahalanobis | 99.23 | 1.68 | 96.89 | 23.27 | 89.01 | 52.26 | 93.75 | 32.28 | 98.81 | 6.44 | 99.29 | 3.13 | 96.16 | 19.84 |
| Residual | 99.05 | 1.86 | 95.61 | 31.96 | 82.22 | 67.74 | 94.48 | 31.92 | 99.19 | 3.05 | 99.36 | 2.03 | 94.98 | 23.09 |
| Vim | 97.23 | 14.33 | 92.88 | 36.41 | 93.82 | 28.66 | 89.94 | 41.40 | 91.58 | 38.73 | 95.13 | 23.97 | 93.43 | 30.58 |
| NECO | 97.67 | 12.20 | 94.04 | 33.31 | 93.57 | 31.58 | 90.25 | 41.08 | 92.65 | 34.50 | 95.90 | 21.27 | 94.02 | 28.99 |
| MCM | 72.98 | 92.09 | 90.75 | 63.39 | 75.53 | 88.66 | 65.54 | 93.36 | 50.79 | 99.11 | 60.97 | 97.79 | 69.43 | 89.06 |
| Trusted | 99.98 | 0.04 | 97.21 | 17.80 | 86.32 | 61.45 | 97.13 | 15.68 | 99.95 | 0.03 | 99.96 | 0.08 | 96.76 | 15.85 |
| KL-matching | 94.32 | 25.12 | 90.69 | 38.25 | 90.69 | 38.52 | 84.16 | 52.94 | 77.85 | 70.96 | 86.99 | 47.80 | 87.45 | 45.60 |
| NNguide | 97.91 | 10.23 | 97.36 | 13.61 | 92.62 | 17.12 | 89.88 | 41.93 | 86.77 | 60.83 | 93.14 | 35.04 | 92.95 | 33.13 |
| RelativeMaha | 96.70 | 14.36 | 96.44 | 19.35 | 91.64 | 46.09 | 86.99 | 51.46 | 82.57 | 61.12 | 91.69 | 34.71 | 91.01 | 37.85 |
| KNN | 97.38 | 13.88 | 96.93 | 16.57 | 91.32 | 46.27 | 90.31 | 39.47 | 86.24 | 60.08 | 93.00 | 34.22 | 92.53 | 35.08 |
| X-Maha (ours) | 99.94 | 0.00 | 98.13 | 8.99 | 88.74 | 52.46 | 97.31 | 13.23 | 99.93 | 0.05 | 99.95 | 0.04 | 97.33 | 12.46 |

Tiny ImageNet [27], LSUN [56], and Places365 [60]. For experiments with ImageNet and ImageNet-LT as the ID datasets, our primary evaluation employs five established OOD datasets: Textures [6], Places365 [60], iNaturalist [47], ImageNet-O [15], and SUN [55]. Extended analysis using OpenOODv1.5 [59] is presented in the Appendix.

**Baselines.** We compare our method with MSP [13], MLS [12], Energy [31], Mahalanobis [28], Residual and Vim [48], NECO [2], MCM [36], Trusted [7], NNguide[38], KNN[46], RelativeMaha[41], and KL-matching [12]. For Mahalanobis, we follow the setting in [10], which uses only the final feature instead of an ensemble of multiple layers [20, 28]. It is worth noting that all these baselines are reimplemented based on our fine-tuned models, except that MCM uses zero-shot CLIP.

**Implementation details.** We implement our approach and *all competing methods* in the same framework on top of the ImageNet-21k pre-trained Vision Transformer (ViT) [8] and the official pre-trained CLIP model. We fine-tune the pre-trained models using in-distribution data for downstream tasks. We employ a batch size of 64 for all experiments. For CIFAR-100 and CIFAR-100-LT, we set the initial learning rate to $0.01$ with a cosine annealing scheduler and fine-tune for $10$ epochs. For ImageNet and ImageNet-LT, the initial learning rate is set to $0.1$, with a cosine annealing scheduler, and the models are fine-tuned for 5 and 20 epochs, respectively. We set $\lambda = 1$ on ImageNet and $\lambda = 0.1$ on CIFAR-100 for the CLIP model to calculate the scoring function. For the Adaptformer module, we set the dimension to $\frac{C}{2L}$, where $C$ is the number of classes, and $L$ is the number of blocks in the ViT model. Other hyperparameters include a momentum of $0.9$, and a weight decay of $5 \times 10^{-4}$, following LIFT [44]. For all baseline methods, we ensure a fair comparison by using the same hyperparameter settings. All experiments are conducted on a single NVIDIA RTX 3090 GPU.

Table 2: Performance on ImageNet (ID) and five OOD datasets. † indicates the results are taken from their papers, except that results for MCM on ImageNet-O are reproduced using official codebase.

| Method | Texture | | Places | | SUN | | iNaturalist | | ImageNet-O | | Average | |
|---|---|---|---|---|---|---|---|---|---|---|---|---|
| | AUROC | FPR95 | AUROC | FPR95 | AUROC | FPR95 | AUROC | FPR95 | AUROC | FPR95 | AUROC | FPR95 |
| IMAGENET-21K PRE-TRAINED VIT | | | | | | | | | | | | |
| MSP | 84.89 | 51.88 | 84.52 | 59.44 | 85.31 | 56.52 | 95.86 | 18.73 | 82.24 | 60.00 | 86.56 | 49.31 |
| MLS | 90.12 | 37.80 | 88.01 | 51.67 | 89.72 | 47.21 | 97.98 | 8.75 | 89.79 | 44.65 | 91.12 | 38.02 |
| Energy | 90.72 | 34.65 | 88.15 | 50.40 | 90.06 | 45.31 | 98.23 | 7.41 | 90.73 | 41.00 | 91.58 | 35.75 |
| Mahalanobis | 92.93 | 26.31 | 89.27 | 47.56 | 91.53 | 39.82 | 99.33 | 2.72 | 92.12 | 37.50 | 93.03 | 30.78 |
| Residual | 92.84 | 30.66 | 84.80 | 61.14 | 88.34 | 50.14 | 98.02 | 9.51 | 87.11 | 52.50 | 90.22 | 40.79 |
| Vim | 91.04 | 33.33 | 88.37 | 49.82 | 90.30 | 44.34 | 98.37 | 6.86 | 90.92 | 40.20 | 91.80 | 34.91 |
| NECO | 92.13 | 30.16 | 89.92 | 46.49 | 91.95 | 40.11 | 98.99 | 4.12 | 91.45 | 39.80 | 92.89 | 32.14 |
| NECO† | 92.86 | 32.44 | **90.38** | **42.66** | **93.15** | **33.98** | 99.34 | 3.26 | **94.53** | **25.20** | 94.05 | 27.51 |
| Trusted | 43.56 | 86.45 | 46.82 | 96.95 | 50.95 | 94.75 | 49.36 | 91.48 | 39.15 | 95.45 | 45.97 | 93.02 |
| KL-matching | 87.85 | 40.92 | 86.76 | 53.02 | 87.89 | 49.19 | 97.84 | 8.84 | 86.25 | 49.20 | 89.32 | 40.23 |
| NNguide | 90.98 | 35.90 | 87.63 | 54.90 | 89.12 | 51.52 | 98.62 | 5.55 | 90.40 | 48.00 | 91.35 | 39.17 |
| RelativeMaha | 90.28 | 39.80 | 88.05 | 52.59 | 89.83 | 47.53 | 99.06 | 3.54 | 90.23 | 46.10 | 91.49 | 37.91 |
| KNN | 89.18 | 42.16 | 85.45 | 64.49 | 85.69 | 66.31 | 98.03 | 9.86 | 87.45 | 60.45 | 89.16 | 48.65 |
| X-Maha (ours) | **96.65** | **11.70** | 89.64 | 46.00 | 92.04 | 37.78 | **99.40** | **2.26** | 93.76 | 29.80 | **94.30** | **25.51** |
| CLIP-VIT-B/16 | | | | | | | | | | | | |
| MSP | 83.05 | 57.59 | 79.83 | 68.39 | 79.33 | 70.29 | 89.74 | 41.95 | 78.60 | 71.00 | 82.11 | 61.84 |
| MLS | 88.76 | 45.43 | 86.02 | 57.05 | 86.39 | 58.28 | 95.57 | 23.45 | 86.53 | 61.15 | 88.65 | 49.07 |
| Energy | 89.26 | 44.01 | 86.59 | 54.39 | 87.12 | 54.85 | **96.38** | **17.67** | **87.32** | **58.30** | 89.33 | 45.84 |
| Mahalanobis | 85.05 | 66.49 | 84.34 | 72.06 | 85.15 | 75.37 | 90.35 | 65.00 | 80.71 | 79.00 | 85.12 | 71.58 |
| Residual | 76.25 | 80.05 | 75.64 | 88.95 | 75.40 | 91.87 | 71.20 | 94.15 | 67.87 | 88.10 | 73.27 | 88.62 |
| Vim | 89.30 | 44.20 | 86.70 | 54.49 | 87.22 | 55.21 | 96.17 | 18.83 | 87.17 | 59.25 | 89.31 | 46.40 |
| NECO | 88.77 | 47.02 | 87.86 | 52.40 | 88.61 | 53.92 | 95.24 | 25.30 | 85.29 | 64.00 | 89.15 | 48.53 |
| MCM† | 86.11 | 57.77 | 89.77 | 44.69 | 92.57 | 37.59 | 94.61 | 30.91 | 79.51 | 75.70 | 88.51 | 49.33 |
| Trusted | **95.87** | **19.80** | 74.59 | 78.06 | 76.71 | 76.42 | 84.61 | 72.77 | 84.12 | 62.40 | 83.18 | 61.89 |
| KL-matching | 86.64 | 46.45 | 83.28 | 59.25 | 83.21 | 61.23 | 94.18 | 24.99 | 83.19 | 62.45 | 86.10 | 50.87 |
| NNguide | 87.60 | 51.05 | 81.94 | 71.29 | 82.98 | 74.77 | 93.14 | 38.88 | 85.27 | 67.85 | 86.19 | 60.77 |
| RelativeMaha | 85.14 | 62.00 | 81.81 | 63.13 | 83.45 | 63.73 | 94.53 | 25.21 | 83.07 | 67.55 | 85.60 | 56.36 |
| KNN | 83.35 | 68.35 | 77.31 | 81.27 | 76.03 | 87.32 | 87.74 | 75.30 | 81.63 | 79.75 | 81.21 | 78.40 |
| X-Maha (ours) | 89.11 | 49.52 | **90.64** | **41.44** | **93.11** | **35.77** | 95.49 | 23.04 | 82.39 | 69.40 | **90.15** | **43.83** |

## 4.2 Main Results

**Result on CIFAR-100.** As shown in Table 1, our proposed method, X-Maha, outperforms state-of-the-art approaches across multiple OOD datasets. In particular, the average performance of X-Maha on both the CLIP model and the ImageNet-21k pre-trained ViT significantly surpasses previous

methods. X-Maha achieves perfect separation of ID and OOD data on Texture, LSUN, and Places365 datasets. However, we observe a decrease in the performance when using CIFAR-10 as the OOD data. This reduction can be attributed to the high similarity between CIFAR-10 and CIFAR-100 in terms of characteristics, resolution, and visual style—both datasets consist of low-resolution, $32 \times 32$ images with somewhat blurred features, making certain samples challenging to differentiate, even for human observers. This resemblance leads to overlapping feature representations in the shallow layers, resulting in relatively diminished performance. Notably, MCM [36] is a zero-shot CLIP-based OOD detection method, and its performance is significantly inferior to other methods, highlighting the necessity of fine-tuning for downstream tasks.

**Result on ImageNet.** Table 2 summarizes the performance of our proposed method, X-Maha, on the ImageNet dataset. Across both pre-trained models, namely, the ImageNet-21k pre-trained ViT and CLIP-ViT-B/16, X-Maha consistently outperforms existing methods. Specifically, when using the ImageNet-21k pre-trained ViT, X-Maha improves the FPR95 by more than 2% on average compared to the second-best method Mahalanobis [28]. Notably, while MCM [36] does not require fine-tuning, it achieves competitive performance across four OOD datasets, except ImageNet-O. Its overall average performance is on par with the Vim [48] and NECO [2] methods. However, X-Maha still outperforms MCM by $\sim 1.5\%$ in AUROC and $\sim 5.5\%$ in FPR95.

**Result on CIFAR-100-LT.** Table 3 presents the results on the long-tailed version of CIFAR-100 dataset. It can be seen that our method consistently outperforms previous approaches. When using the CLIP model, our method effectively reduces the FPR95 by an average of 6.31% (from 23.56% to 17.25%).

Table 3: OOD detection performance on CIFAR-100-LT (ID) and six OOD datasets.

| Method | Texture | | SVHN | | CIFAR10 | | Tiny ImageNet | | LSUN | | Places365 | | Average | |
| --- | --- | --- | --- | --- | --- | --- | --- | --- | --- | --- | --- | --- | --- | --- |
| | AUROC | FPR95 | AUROC | FPR95 | AUROC | FPR95 | AUROC | FPR95 | AUROC | FPR95 | AUROC | FPR95 | AUROC | FPR95 |
| IMAGENET-21K PRE-TRAINED VIT | | | | | | | | | | | | | | |
| MSP | 97.21 | 13.12 | 95.52 | 24.13 | 91.92 | 38.50 | 85.27 | 48.02 | 84.06 | 64.81 | 90.47 | 43.10 | 90.75 | 38.61 |
| MLS | 99.83 | 0.62 | 96.38 | 18.35 | 94.94 | 25.58 | 90.36 | 34.08 | 98.58 | 7.52 | 99.26 | 3.06 | 96.56 | 14.87 |
| Energy | 99.89 | 0.43 | 95.65 | 24.00 | 94.49 | 29.42 | 90.38 | 34.32 | 99.09 | 4.00 | 99.52 | 1.62 | 96.50 | 15.63 |
| Mahalanobis | 99.96 | 0.20 | 99.33 | 2.51 | **95.09** | 25.98 | 97.63 | 9.26 | 99.48 | 2.26 | 99.57 | 1.71 | _98.51_ | _6.99_ |
| Residual | 99.98 | 0.05 | 97.33 | 17.74 | 86.41 | 62.76 | 98.52 | 6.90 | 99.83 | 0.47 | 99.80 | 0.45 | 96.98 | 14.73 |
| Vim | 99.91 | 0.28 | 96.18 | 20.72 | 94.56 | 29.01 | 91.27 | 31.59 | 99.25 | 3.20 | 99.60 | 1.23 | 96.80 | 14.34 |
| NECO | 99.86 | 0.64 | 97.37 | 13.58 | 94.91 | **24.62** | 91.22 | 29.21 | 98.39 | 10.21 | 99.22 | 3.78 | 96.83 | 13.67 |
| Trusted | 100.0 | 0.00 | 99.12 | 3.60 | 87.34 | 52.84 | 97.67 | 10.37 | 99.97 | 0.00 | 99.98 | 0.00 | 97.35 | 11.13 |
| KL-matching | 98.48 | 6.40 | 97.44 | 12.11 | 94.00 | 26.88 | 87.56 | 38.91 | 86.65 | 52.78 | 92.94 | 31.01 | 92.84 | 28.02 |
| KL-matching | 98.48 | 6.40 | 97.44 | 12.11 | 94.00 | 26.88 | 87.56 | 38.91 | 86.65 | 52.78 | 92.94 | 31.01 | 92.84 | 28.02 |
| NNguide | 99.19 | 3.30 | 98.89 | 3.16 | 95.10 | 25.71 | 90.95 | 30.58 | 92.92 | 34.72 | 95.60 | 23.14 | 95.44 | 20.10 |
| RelativeMaha | 97.25 | 11.19 | 96.01 | 26.31 | 94.65 | 26.99 | 89.94 | 45.41 | 90.46 | 53.53 | 92.43 | 38.72 | 93.46 | 33.69 |
| KNN | 98.82 | 4.57 | 97.44 | 14.68 | 92.51 | 36.42 | 88.45 | 37.86 | 89.78 | 43.25 | 94.14 | 28.30 | 93.52 | 27.51 |
| X-Maha (ours) | **100.0** | **0.00** | **99.75** | **0.43** | 94.22 | 29.86 | **99.75** | **1.12** | **99.99** | **0.00** | **99.99** | 0.01 | **98.95** | **5.24** |
| CLIP-VIT-B/16 | | | | | | | | | | | | | | |
| MSP | 91.05 | 39.34 | 86.13 | 48.73 | 85.33 | 55.47 | 78.22 | 68.10 | 73.52 | 76.50 | 83.16 | 57.92 | 82.90 | 57.68 |
| MLS | 96.76 | 16.95 | 88.44 | 49.78 | 91.85 | 36.84 | 87.05 | 47.53 | 90.35 | 36.47 | 94.29 | 25.52 | 91.46 | 35.57 |
| Energy | 97.31 | 13.09 | 86.40 | 59.64 | **92.37** | **34.15** | 88.01 | 43.79 | 92.25 | 28.45 | 95.49 | 19.49 | 91.97 | 33.10 |
| Mahalanobis | 99.11 | 1.03 | 95.92 | 29.87 | 84.76 | 60.58 | 90.97 | 43.83 | 99.08 | 4.07 | 99.28 | 1.99 | 94.85 | 23.56 |
| Residual | 98.90 | 1.42 | 94.83 | 33.99 | 77.19 | 73.51 | 91.24 | 48.57 | 99.28 | 1.94 | 99.34 | 0.87 | 93.46 | 26.72 |
| Vim | 98.12 | 9.17 | 88.61 | 52.92 | 92.19 | 35.68 | 88.97 | 41.63 | 94.26 | 21.87 | 96.76 | 14.76 | 93.15 | 29.34 |
| NECO | 98.00 | 9.57 | 91.32 | 41.13 | 91.11 | 40.21 | 87.51 | 46.54 | 93.99 | 23.37 | 96.71 | 16.22 | 93.11 | 29.51 |
| Trusted | **99.97** | 0.11 | 93.57 | 43.80 | 80.76 | 70.36 | **95.46** | **25.58** | 99.95 | 0.10 | **99.95** | 0.08 | _94.94_ | _23.34_ |
| KL-matching | 95.01 | 21.76 | 90.76 | 31.69 | 88.87 | 44.17 | 81.68 | 57.93 | 79.31 | 63.65 | 87.64 | 43.21 | 87.21 | 43.73 |
| NNguide | 96.24 | 15.50 | 95.95 | 20.93 | 87.06 | 57.99 | 84.36 | 60.30 | 88.23 | 48.69 | 91.11 | 32.20 | 90.49 | 39.27 |
| RelativeMaha | 97.25 | 11.19 | 96.01 | 26.31 | 94.65 | 26.99 | 89.94 | 45.41 | 90.46 | 53.53 | 92.43 | 38.72 | 93.46 | 33.69 |
| KNN | 97.48 | 12.68 | 91.39 | 44.15 | 84.87 | 59.53 | 84.54 | 53.89 | 83.62 | 56.77 | 90.25 | 41.52 | 88.69 | 44.76 |
| X-Maha (ours) | 99.90 | **0.02** | 97.41 | 16.80 | 85.35 | 59.44 | 94.85 | 27.10 | 99.89 | **0.10** | 99.91 | **0.03** | 96.22 | 17.25 |

**Result on ImageNet-LT.** Additionally, Table 4 presents the results on the long-tailed ImageNet dataset. It can be seen that our method consistently outperforms previous approaches. On average, our method reduces FPR95 by 5.29% and 2.87% for ImageNet-21k pre-trained ViT and CLIP, respectively. The AUROC also improves by 1.77% when using the CLIP model.

## 4.3 Ablation Studies

**Why X-Maha works?** Unless otherwise specified, in this subsection, we use the ImageNet-21k pre-trained ViT as the default base model. Figure 4 presents a comparison of OOD score distributions with and without the application of our proposed X-Maha method. When X-Maha is not applied, only the final layer features are used to compute the Mahalanobis distance as a scoring function.

Table 4: OOD detection performance on ImageNet-LT (ID) and five OOD datasets.

| Method | Texture | | Places | | SUN | | iNaturalist | | ImageNet-O | | **Average** | |
|---|---|---|---|---|---|---|---|---|---|---|---|---|
| | AUROC | FPR95 | AUROC | FPR95 | AUROC | FPR95 | AUROC | FPR95 | AUROC | FPR95 | AUROC | FPR95 |
| IMAGENET-21K PRE-TRAINED VIT | | | | | | | | | | | | |
| MSP | 86.04 | 47.50 | 85.20 | 56.52 | 86.36 | 53.07 | 97.17 | 11.97 | 83.68 | 57.40 | 87.69 | 45.29 |
| MLS | 90.18 | 38.71 | 88.76 | 49.34 | 90.39 | 45.44 | 98.47 | 6.71 | 88.91 | 47.90 | 91.34 | 37.62 |
| Energy | 90.87 | 35.51 | 89.29 | 45.97 | 91.05 | 41.04 | 98.78 | 5.06 | 89.83 | 42.80 | 91.96 | 34.08 |
| Mahalanobis | 92.99 | 26.95 | 89.48 | 46.34 | 91.71 | 38.35 | 99.28 | 2.84 | 91.66 | 38.85 | 93.02 | 30.67 |
| Residual | 91.60 | 35.74 | 82.23 | 65.71 | 86.58 | 55.54 | 97.44 | 12.67 | 84.05 | 59.05 | 88.38 | 45.74 |
| Vim | 91.23 | 34.10 | 89.47 | 45.23 | 91.27 | 39.83 | 98.88 | 4.77 | 90.05 | 41.70 | 92.18 | 33.13 |
| NECO | 91.66 | 31.44 | 89.21 | **43.71** | 91.44 | 37.07 | 98.93 | 4.09 | 89.64 | 42.70 | 92.18 | 31.80 |
| Trusted | 91.98 | 32.36 | 82.11 | 66.31 | 85.72 | 58.34 | 98.09 | 9.29 | 90.91 | 40.15 | 89.76 | 41.29 |
| KL-matching | 88.72 | 38.71 | 87.41 | 50.03 | 89.14 | 45.83 | 98.44 | 6.19 | 87.24 | 47.50 | 90.19 | 37.65 |
| NNguide | 91.06 | 35.85 | 87.66 | 52.60 | 89.53 | 48.61 | 98.66 | 5.33 | 90.08 | 46.55 | 91.40 | 37.79 |
| RelativeMaha | 89.97 | 40.74 | 88.22 | 52.89 | 89.92 | 48.20 | 98.97 | 4.01 | 89.67 | 47.40 | 91.35 | 38.65 |
| KNN | 89.80 | 39.86 | 85.86 | 59.61 | 87.30 | 56.79 | 98.21 | 7.43 | 87.97 | 56.10 | 89.83 | 43.96 |
| X-Maha (ours) | **96.92** | **11.79** | **89.82** | 45.36 | **92.18** | 36.16 | **99.33** | **2.51** | **93.46** | 31.10 | **94.34** | **25.38** |
| CLIP-VIT-B/16 | | | | | | | | | | | | |
| MSP | 81.55 | 60.34 | 79.32 | 65.16 | 78.44 | 66.53 | 90.60 | 38.49 | 78.37 | 71.60 | 81.66 | 60.42 |
| MLS | 87.00 | 52.27 | 85.31 | 56.20 | 85.47 | 57.19 | 95.03 | 25.21 | 84.33 | 65.10 | 87.43 | 51.19 |
| Energy | 87.81 | 50.07 | 86.37 | 51.85 | 86.76 | 53.08 | **95.94** | **19.61** | **85.12** | 63.65 | 88.40 | 47.65 |
| Mahalanobis | 83.81 | 67.64 | 84.44 | 66.85 | 85.50 | 69.58 | 87.49 | 72.57 | 78.82 | 80.20 | 84.01 | 71.37 |
| Residual | 74.81 | 80.71 | 75.62 | 86.49 | 76.56 | 87.93 | 63.27 | 96.67 | 64.43 | 89.30 | 70.94 | 88.22 |
| Vim | 87.90 | 49.72 | 86.52 | 51.32 | 86.96 | 52.47 | 95.55 | 21.06 | 84.96 | 63.90 | 88.38 | 47.69 |
| NECO | 86.67 | 53.67 | 86.71 | 53.11 | 87.17 | 54.63 | 94.08 | 29.95 | 82.90 | 67.60 | 87.51 | 51.79 |
| Trusted | 71.96 | 70.46 | 44.51 | 97.89 | 49.78 | 97.77 | 49.44 | 98.59 | 48.79 | 89.05 | 52.90 | 90.75 |
| KL-matching | 85.35 | 51.56 | 82.84 | 57.00 | 82.51 | 57.56 | 94.54 | 23.36 | 82.52 | 64.00 | 85.55 | 50.70 |
| NNguide | 86.06 | 56.44 | 82.53 | 65.49 | 83.68 | 66.66 | 90.59 | 47.29 | 83.07 | 71.15 | 85.18 | 61.41 |
| RelativeMaha | 83.81 | 67.38 | 81.84 | 63.31 | 82.74 | 64.13 | 93.95 | 31.74 | 82.25 | 71.10 | 84.92 | 59.53 |
| KNN | 82.58 | 66.42 | 78.66 | 74.60 | 78.64 | 77.96 | 83.97 | 70.68 | 79.61 | 79.10 | 80.69 | 73.75 |
| X-Maha (ours) | **89.94** | **46.79** | **90.47** | **42.51** | **92.71** | **37.95** | 94.79 | 27.57 | 82.94 | 69.10 | **90.17** | **44.78** |

Figure 4: Comparisons of OOD score distribution before and after applying our X-Maha method. CIFAR-100 is used as the ID dataset and the OOD dataset from left to right is Texture, Tiny ImageNet, LSUN, and Places365. The horizontal axis represents the OOD score (small values indicate a high likelihood of being OOD samples).

It can be seen that the score distributions for ID samples remain largely consistent, whether or not the X-Maha method is applied. However, the use of X-Maha causes a significant leftward shift in the score distribution for OOD samples. This shift occurs because the features in the final layer of unseen OOD samples are not effectively captured. Furthermore, re-weighted information from the shallow layers amplifies this shift, resulting in better discrimination. As a result, the X-Maha method enhances the separation between ID and OOD samples in the embedding space. This improvement is critical for more accurate identification and differentiation of ID and OOD samples, thus boosting the overall performance and reliability of the detection process.

**Importance weights of each layer.** As depicted in Figure 5, our proposed method can adaptively assign importance weights to different layers. Overall, the first 6 layers are assigned relatively lower weights compared to the rest of the Transformer layers. Notably, the final layer's weight is particularly prominent. This is because the last layer of the feature extractor learns the most discriminative features for in-distribution classes and is important for OOD detection. As shown in the figure, rather than relying solely on the penultimate layer's features, our method effectively utilizes shallow layer features as well.

**Impact of features from shallow layers.** Figure 6 illustrates the effect of fusing features from varying numbers of layers. The x-axis represents the number of layers counted from the penultimate

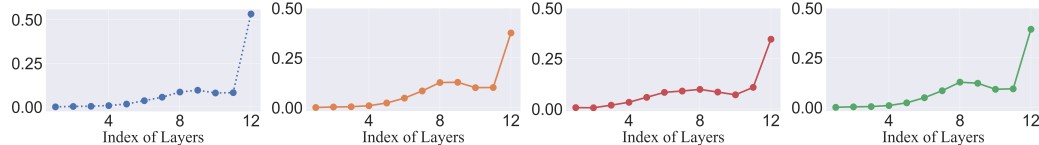

Figure 5: Distribution of layer-specific weights for CIFAR-100, ImageNet, ImageNet (CLIP), and ImageNet-LT where the y-axis denotes AUROC (%).

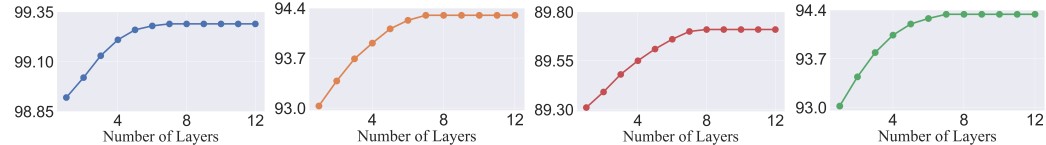

Figure 6: Impact of the number of layers used for feature fusion on OOD detection performance. The ID dataset from left to right is CIFAR-100, ImageNet, ImageNet (CLIP), and ImageNet-LT, where the vertical axis represents AUROC.

layer towards the first, while the y-axis indicates the average OOD detection AUROC. As shown in the figures, using only the penultimate layer's features yields decent results, but fusing the last 6 layers of the Transformer achieves the best performance, highlighting the importance of shallow features. For features from the sixth layer and beyond, their impact on the results is minimal. As discussed in the previous analysis, our method assigns lower weights to these layers accordingly.

Table 5: Comparisons of different feature mixing strategies. 'In21k' denotes ViT pre-trained on ImageNet-21k.

| Method | CIFAR-100 | | | | ImageNet | | | | Average | |
| | CLIP | | In21k | | CLIP | | In21k | | | |
| | AUROC | FPR95 | AUROC | FPR95 | AUROC | FPR95 | AUROC | FPR95 | AUROC | FPR95 |
|---|---|---|---|---|---|---|---|---|---|---|
| Trusted | 96.76 | 15.85 | 98.37 | 8.17 | 83.18 | 61.89 | 45.97 | 93.02 | 81.49 | 44.73 |
| SA | 96.53 | 13.19 | 98.77 | 6.58 | 82.68 | 64.59 | 94.01 | 27.62 | 93.00 | 28.00 |
| PM | 96.15 | 18.13 | 98.16 | 10.05 | 81.03 | 77.59 | 81.86 | 27.47 | 89.30 | 33.31 |
| Flatten12 | 42.10 | 89.67 | 29.00 | 90.93 | - | - | - | - | 34.05 | 90.15 |
| Flatten6 | 93.31 | 15.99 | 81.75 | 49.33 | - | - | - | - | 87.53 | 32.66 |
| Ours | **97.33** | **12.46** | **99.29** | **3.76** | **90.15** | **43.83** | **94.30** | **25.51** | **95.27** | **21.39** |

**Ways to fuse shallow features.** We compare our proposed feature mixing method with other fusion strategies including 1) Trusted [7] which directly employs the arithmetic mean to fuse features from each layer during both the training and test phases; 2) Score Aggregation (SA) [28] which calculates the OOD score via Mahalanobis distance using features from each layer separately and weighted them together. Since SA requires a validation set containing both ID and OOD data, we use the weights derived from our method to calculate the weighted sum of scores; 3) Power Mean (PM) [42] proposes to reweight each layer's feature based on feature norms; 4) Flatten12 concatenates all layers' features into a single vector, while Flatten6 concatenates the last six layers' features. The results are presented in Table 5. It can be seen that our proposed adaptive fusion method achieves a significant advantage in aggregating shallow features, further confirming its effectiveness.

## 5 Conclusion

This paper introduces a timely improvement to Mahalanobis-based OOD detection by effectively mixing Transformer features across layers. While shallow features may lack class discrimination, we demonstrate their strength in separating ID and OOD data. Our method assigns importance weights to layer features, without relying on validation data, and leverages parameter-efficient fine-tuning to better adapt pre-trained Transformers for OOD detection. Extensive experiments validate our approach across zero-shot and fine-tuning settings, vision-only and vision-language models, and both balanced and long-tailed ID datasets. Ablation studies further clarify its mechanisms. We believe this work establishes a strong baseline for future OOD detection research.

## Acknowledgments and Disclosure of Funding

This work was supported by the National Science Foundation of China (62206049, 62225602), and the Big Data Computing Center of Southeast University. We would like to thank anonymous reviewers for their constructive suggestions.

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

# A  Additional Experiments

## A.1  Zero-shot OOD detection performance

To further demonstrate the effectiveness of our proposed feature mixing approach, we evaluate the zero-shot performance without fine-tuning pre-trained models using ID datasets. The results are reported in Table 6.

Table 6: OOD detection performance on ImageNet-LT (ID) and five OOD datasets using the pre-trained models without fine-tuning.

| Method | Texture AUROC | Texture FPR95 | Places AUROC | Places FPR95 | SUN AUROC | SUN FPR95 | iNaturalist AUROC | iNaturalist FPR95 | ImageNet-O AUROC | ImageNet-O FPR95 | Average AUROC | Average FPR95 |
|---|---|---|---|---|---|---|---|---|---|---|---|---|
| **IMAGENET-21K PRE-TRAINED VIT** | | | | | | | | | | | | |
| MSP | 84.92 | 50.12 | 81.63 | 68.06 | 81.77 | 68.88 | 95.02 | 24.12 | 80.04 | 64.65 | 84.68 | 55.17 |
| MLS | 91.01 | 36.22 | **84.48** | **61.71** | **86.96** | 56.93 | 96.88 | 17.52 | 89.31 | 52.45 | 89.73 | 44.97 |
| Energy | 91.17 | 36.44 | 84.11 | 62.57 | 86.91 | 56.66 | 96.30 | 22.56 | 89.72 | 51.90 | 89.64 | 46.03 |
| Mahalanobis | 93.13 | 24.63 | 81.79 | 67.52 | 85.16 | 61.30 | 98.63 | 5.59 | 91.26 | 41.30 | 89.99 | 40.07 |
| Residual | 91.78 | 33.83 | 64.01 | 86.88 | 71.16 | 80.23 | 94.44 | 27.58 | 83.22 | 58.95 | 80.92 | 57.49 |
| Vim | 91.51 | 34.47 | 83.98 | 62.44 | 86.90 | **56.37** | 96.57 | 20.37 | 89.94 | 50.15 | 89.78 | 44.76 |
| NECO | 92.54 | 27.00 | 79.76 | 65.31 | 83.77 | 60.32 | 98.55 | 5.37 | 90.27 | 43.50 | 88.98 | 40.30 |
| KL-matching | 87.51 | 40.66 | 83.23 | 64.05 | 84.18 | 61.98 | 96.72 | 16.12 | 83.40 | 55.15 | 87.01 | 47.59 |
| X-Maha (ours) | **97.15** | **10.20** | 81.51 | 67.20 | 85.49 | 56.49 | **98.69** | **5.13** | **93.15** | **32.15** | **91.20** | **34.23** |
| **CLIP-VIT-B/16** | | | | | | | | | | | | |
| MSP | 76.67 | 82.07 | 61.22 | 94.21 | 61.62 | 96.00 | 72.78 | 92.94 | 70.56 | 89.95 | 68.57 | 91.03 |
| MLS | 76.88 | 91.90 | 78.12 | 85.99 | 76.29 | 93.61 | 74.86 | 95.57 | 74.28 | 89.20 | 76.09 | 91.25 |
| Energy | 69.74 | 97.02 | 78.19 | 89.82 | 75.68 | 96.23 | 69.13 | 98.24 | 69.40 | 92.55 | 72.43 | 94.77 |
| Mahalanobis | 69.91 | 94.86 | 70.20 | 96.58 | 67.53 | 98.78 | 67.77 | 98.96 | 70.57 | 90.30 | 69.20 | 95.90 |
| Residual | 64.57 | 96.51 | 59.59 | 98.11 | 55.23 | 99.38 | 49.12 | 99.35 | 62.89 | 91.65 | 58.28 | 97.00 |
| Vim | 69.34 | 96.99 | 76.89 | 91.80 | 74.19 | 97.36 | 67.49 | 98.44 | 68.96 | 92.10 | 71.37 | 95.34 |
| NECO | 73.70 | 92.59 | 73.55 | 93.38 | 71.99 | 97.32 | 70.80 | 98.68 | 70.96 | 89.65 | 72.20 | 94.32 |
| KL-matching | 83.71 | 63.99 | 62.27 | 86.25 | 63.80 | 88.92 | 79.40 | 79.31 | 74.73 | 80.90 | 72.78 | 79.87 |
| X-Maha (ours) | **87.74** | **58.03** | **88.22** | **59.63** | **89.77** | **61.21** | **91.07** | **55.93** | **80.75** | **76.60** | **87.51** | **62.28** |

## A.2  In-distribution classification accuracy

Our fine-tuned model also shows strong ID classification performance, as detailed in Table 7. In terms of overall accuracy, both CIFAR-100 and ImageNet-1k perform better with balanced data compared to long-tailed data. This indicates that data balance positively impacts model performance, facilitating more accurate classification tasks.

When comparing different models, the pre-trained ViT consistently outperform CLIP-ViT-B/16 in most scenarios. This indicates that the pre-trained ViT has specific advantages for these data sets and tasks, suggesting that its pre-training approach is more suitable for these classification tasks, thereby also enhancing its efficacy in OOD detection tasks.

Table 7: Top 1% accuracy on ID data for the original classification task, for the models.

| ID dataset | Label distribution | Model | Accuracy (%) |
|---|---|---|---|
| CIFAR-100 | Zero-shot | CLIP-ViT-B/16 | 66.69 |
| | Long-tailed | CLIP-ViT-B/16 | 82.87 |
| | | Pre-trained ViT | 89.99 |
| | Balanced | CLIP-ViT-B/16 | 88.59 |
| | | Pre-trained ViT | 93.47 |
| ImageNet-1k | Zero-shot | CLIP-ViT-B/16 | 67.12 |
| | Long-tailed | CLIP-ViT-B/16 | 75.82 |
| | | Pre-trained ViT | 81.79 |
| | Balanced | CLIP-ViT-B/16 | 79.08 |
| | | Pre-trained ViT | 83.50 |

Table 8: Ablation studies on weights of different layers on CIFAR-100 (ID).

| Method | Texture AUROC | FPR95 | SVHN AUROC | FPR95 | CIFAR10 AUROC | FPR95 | Tiny ImageNet AUROC | FPR95 | LSUN AUROC | FPR95 | Places365 AUROC | FPR95 | Average AUROC | FPR95 |
|---|---|---|---|---|---|---|---|---|---|---|---|---|---|---|
| | | | | | | **IMAGENET-21K PRE-TRAINED VIT** | | | | | | | | |
| $W_{0.083}$ | 100.0 | 0.00 | **99.61** | **0.28** | 90.02 | 51.14 | **100.0** | **0.00** | 100.0 | 0.00 | 100.0 | 0.00 | 98.27 | 8.57 |
| $W_{0.5}$ | 100.0 | 0.00 | 99.43 | 2.36 | 96.76 | 18.26 | 99.62 | 1.81 | 100.0 | 0.00 | 99.99 | 0.01 | **99.30** | **3.74** |
| $W_{0.75}$ | 99.99 | 0.04 | 99.27 | 3.33 | 96.99 | 16.82 | 98.83 | 5.22 | 99.89 | 0.06 | 99.89 | 0.26 | 99.14 | 4.29 |
| $W_{1.0}$ | 99.97 | 0.12 | 99.16 | 3.92 | **97.09** | **16.49** | 97.99 | 8.96 | 99.61 | 1.07 | 99.67 | 1.33 | 98.92 | 5.32 |
| X-Maha (ours) | 100.0 | 0.00 | 99.50 | 1.91 | 96.47 | 19.52 | 99.80 | 1.11 | 100.0 | 0.00 | 100.0 | 0.00 | 99.29 | 3.76 |
| | | | | | | **CLIP-VIT-B/16** | | | | | | | | |
| $W_{0.083}$ | 100.0 | 0.00 | **99.05** | **2.82** | 83.92 | 65.74 | **99.94** | **0.08** | 100.0 | 0.00 | 100.0 | 0.00 | 97.15 | 11.44 |
| $W_{0.5}$ | 99.92 | 0.07 | 98.00 | 10.27 | 89.15 | 51.09 | 96.49 | 17.36 | 99.88 | 0.21 | 99.91 | 0.12 | 97.22 | 13.19 |
| $W_{0.75}$ | 99.61 | 0.85 | 97.84 | 12.34 | **89.36** | **50.66** | 94.38 | 26.56 | 99.24 | 3.85 | 99.55 | 1.68 | 96.66 | 15.99 |
| $W_{1.0}$ | 99.23 | 1.68 | 96.89 | 23.27 | 89.01 | 52.26 | 93.75 | 32.28 | 98.81 | 6.44 | 99.29 | 3.13 | 96.16 | 19.84 |
| X-Maha (ours) | 99.95 | 0.02 | 98.31 | 8.62 | 88.56 | 53.97 | 97.54 | 12.91 | 99.93 | 0.06 | 99.95 | 0.02 | 97.37 | 12.60 |

## A.3 Ablation studies on weights of different layers

To further emphasize the importance of differentiated layer weighting, we provide experimental tables (i.e., Table 8, 9, 10). In these Tables, we test different scenarios where the final layer is given weights of 0.083 (i.e., uniform), 0.5, 0.75, and 1 (which are represented by $W_{0.083}, W_{0.5}, W_{0.75}, W_{1.0}$), while the other layers receive the remaining weights evenly. Overall, the OOD detection performance is sensitive to layer weights; however, our X-Maha approach consistently achieves remarkable performance.

Table 9: Ablation studies on weights of different layers on CIFAR-100-LT (ID).

| Method | Texture AUROC | FPR95 | SVHN AUROC | FPR95 | CIFAR10 AUROC | FPR95 | Tiny ImageNet AUROC | FPR95 | LSUN AUROC | FPR95 | Places365 AUROC | FPR95 | Average AUROC | FPR95 |
|---|---|---|---|---|---|---|---|---|---|---|---|---|---|---|
| | | | | | | **IMAGENET-21K PRE-TRAINED VIT** | | | | | | | | |
| $W_{0.083}$ | 100.0 | 0.00 | **99.85** | **0.01** | 85.54 | 68.56 | **100.0** | **0.00** | 100.0 | 0.00 | 100.0 | 0.00 | 97.57 | 11.43 |
| $W_{0.5}$ | 100.0 | 0.00 | 99.68 | 0.68 | 94.56 | 27.90 | 99.58 | 1.68 | 99.99 | 0.00 | 99.98 | 0.01 | **98.97** | **5.05** |
| $W_{0.75}$ | 99.99 | 0.05 | 99.49 | 1.66 | 94.93 | 26.42 | 98.67 | 5.57 | 99.81 | 0.39 | 99.83 | 0.53 | 98.79 | 5.77 |
| $W_{1.0}$ | 99.96 | 0.20 | 99.33 | 2.51 | **95.09** | **25.98** | 97.63 | 9.26 | 99.48 | 2.26 | 99.57 | 1.71 | 98.51 | 6.99 |
| X-Maha (ours) | **100.0** | **0.00** | 99.75 | 0.43 | 94.22 | 29.86 | 99.75 | 1.12 | 99.99 | 0.00 | 99.99 | 0.01 | 98.95 | 5.24 |
| | | | | | | **CLIP-VIT-B/16** | | | | | | | | |
| $W_{0.083}$ | 100.0 | 0.00 | **98.76** | **5.09** | 80.77 | 68.39 | **99.89** | **0.20** | 100.0 | 0.00 | 100.0 | 0.00 | 96.57 | 12.28 |
| $W_{0.5}$ | 99.88 | 0.04 | 97.37 | 16.90 | 85.52 | 58.97 | 94.36 | 28.65 | 99.87 | 0.11 | 99.89 | 0.05 | 96.15 | 17.45 |
| $W_{0.75}$ | 99.52 | 0.55 | 97.15 | 19.66 | 85.73 | 58.99 | 91.58 | 39.22 | 99.36 | 2.63 | 99.52 | 1.20 | 95.48 | 20.38 |
| $W_{1.0}$ | 99.11 | 1.03 | 95.92 | 29.87 | 84.76 | 60.58 | 90.97 | 43.83 | 99.08 | 4.07 | 99.28 | 1.99 | 94.85 | 23.50 |
| X-Maha (ours) | 99.94 | 0.00 | 98.13 | 8.99 | **88.74** | **52.46** | 97.31 | 13.23 | 99.93 | 0.05 | 99.95 | 0.04 | 97.33 | 12.46 |

Table 10: Ablation studies on weights of different layers on ImageNet-1k-LT (ID).

| Method | Texture AUROC | FPR95 | Places AUROC | FPR95 | SUN AUROC | FPR95 | iNaturalist AUROC | FPR95 | ImageNet-O AUROC | FPR95 | Average AUROC | FPR95 |
|---|---|---|---|---|---|---|---|---|---|---|---|---|
| | | | | | **IMAGENET-21K PRE-TRAINED VIT** | | | | | | | |
| $W_{0.083}$ | **98.55** | **6.45** | 86.32 | 60.57 | 88.92 | 49.25 | 98.02 | 9.42 | 91.72 | 37.05 | 92.71 | 32.55 |
| $W_{0.5}$ | 95.02 | 17.96 | 89.76 | **44.79** | 92.08 | 36.60 | **99.36** | 2.63 | 92.68 | 34.30 | 93.78 | 27.26 |
| $W_{0.75}$ | 93.78 | 23.39 | 89.62 | 45.54 | 91.88 | 37.22 | 99.32 | 2.74 | 92.06 | 36.75 | 93.33 | 29.13 |
| $W_{1.0}$ | 92.99 | 26.95 | 89.48 | 46.34 | 91.71 | 38.35 | 99.28 | 2.84 | 91.66 | 38.85 | 93.02 | 30.67 |
| X-Maha (ours) | 96.92 | 11.79 | 89.82 | 45.36 | **92.18** | **36.16** | 99.33 | **2.51** | **93.46** | **31.10** | **94.34** | **25.38** |
| | | | | | **CLIP-VIT-B/16** | | | | | | | |
| $W_{0.083}$ | **92.23** | **36.76** | 91.11 | 39.65 | **93.02** | **36.38** | 94.62 | 29.54 | **83.30** | **67.45** | 90.86 | 41.96 |
| $W_{0.5}$ | 88.52 | 52.23 | 89.87 | 45.47 | 92.21 | 40.58 | 94.54 | 28.45 | 82.56 | 71.00 | 89.54 | 47.55 |
| $W_{0.75}$ | 87.68 | 55.39 | 89.69 | 46.37 | 92.08 | 41.23 | 94.41 | 29.30 | 82.23 | 72.40 | 89.22 | 48.94 |
| $W_{1.0}$ | 83.81 | 67.64 | 84.44 | 66.85 | 85.50 | 69.58 | 87.49 | 72.57 | 78.82 | 80.20 | 84.01 | 71.37 |
| X-Maha (ours) | 89.94 | 46.79 | **90.47** | 42.51 | 92.71 | 37.95 | **94.79** | **27.57** | 82.94 | 69.10 | 90.17 | 44.78 |

## A.4 Ablation studies on smaller pre-trained transformers

As depicted in Table 11, 12, 13, and 14, we have included models like vit_tiny_patch16_224 and vit_small_patch16_224, shown in the upper and lower sections of each table. The outcomes from these smaller models provide further confirmation that our OOD score remains robust and effective across various model scales, thereby enhancing the generalizability and reliability of our proposed approach.

Table 11: OOD detection performance on CIFAR-100 (ID) on smaller transformers.

| Method | Texture | | SVHN | | CIFAR10 | | Tiny ImageNet | | LSUN | | Places365 | | Average | |
|---|---|---|---|---|---|---|---|---|---|---|---|---|---|---|
| | AUROC | FPR95 | AUROC | FPR95 | AUROC | FPR95 | AUROC | FPR95 | AUROC | FPR95 | AUROC | FPR95 | AUROC | FPR95 |
| VIT_TINY_PATCH16_224 | | | | | | | | | | | | | | |
| MSP | 92.09 | 35.34 | 83.28 | 61.28 | 83.30 | 63.73 | 79.89 | 69.73 | 72.86 | 84.07 | 82.13 | 65.55 | 82.26 | 63.28 |
| MLS | 98.62 | 6.13 | 92.09 | 35.85 | 87.39 | 54.92 | 87.71 | 52.79 | 88.92 | 57.57 | 94.27 | 30.32 | 91.50 | 39.60 |
| Energy | 99.03 | 4.26 | 92.78 | 32.41 | 87.28 | 55.96 | 88.18 | 51.27 | 90.32 | 51.04 | 95.25 | 24.94 | 92.14 | 36.65 |
| Mahalanobis | 99.90 | 0.35 | 96.28 | 15.78 | 87.78 | 56.67 | 92.48 | 33.81 | 98.20 | 9.10 | 98.77 | 6.27 | 95.57 | 20.33 |
| Residual | 99.71 | 0.85 | 86.24 | 52.70 | 76.86 | 72.72 | 90.89 | 42.55 | 97.46 | 14.02 | 97.25 | 13.66 | 91.40 | 32.75 |
| Vim | 99.19 | 3.62 | 92.99 | 31.18 | 87.49 | 54.90 | 88.70 | 48.96 | 91.14 | 47.63 | 95.67 | 22.63 | 92.53 | 34.82 |
| NECO | 99.17 | 3.83 | 92.34 | 34.24 | 87.85 | 53.47 | 89.47 | 46.45 | 92.38 | 43.05 | 96.06 | 21.11 | 92.88 | 33.69 |
| KL-matching | 95.41 | 18.40 | 87.58 | 45.71 | 86.32 | 53.75 | 82.47 | 60.45 | 75.28 | 79.24 | 85.65 | 52.82 | 85.45 | 51.73 |
| X-Maha (ours) | 100.0 | 0.02 | 96.85 | 14.13 | 86.48 | 60.56 | 97.00 | 15.34 | 99.98 | 0.01 | 99.96 | 0.10 | 96.71 | 15.03 |
| VIT_SMALL_PATCH16_224 | | | | | | | | | | | | | | |
| MSP | 95.98 | 19.17 | 92.29 | 38.18 | 90.82 | 39.01 | 85.95 | 52.36 | 82.84 | 68.92 | 89.31 | 47.87 | 89.53 | 44.25 |
| MLS | 99.28 | 3.16 | 96.35 | 18.16 | 95.22 | 24.90 | 92.18 | 32.72 | 96.21 | 25.40 | 97.71 | 13.44 | 96.16 | 19.63 |
| Energy | 99.48 | 2.29 | 96.54 | 16.55 | 95.42 | 23.24 | 92.59 | 29.99 | 97.12 | 18.57 | 98.25 | 10.09 | 96.57 | 16.79 |
| Mahalanobis | 99.91 | 0.59 | 99.05 | 4.72 | 94.65 | 28.65 | 97.53 | 11.36 | 99.54 | 1.78 | 99.54 | 2.52 | 98.38 | 8.27 |
| Residual | 99.96 | 0.11 | 98.60 | 7.06 | 88.66 | 52.27 | 98.09 | 9.68 | 99.65 | 0.75 | 99.67 | 1.14 | 97.44 | 11.83 |
| Vim | 99.56 | 1.99 | 96.88 | 14.63 | 95.46 | 23.06 | 93.17 | 27.68 | 97.52 | 16.28 | 98.47 | 8.82 | 96.84 | 15.41 |
| NECO | 99.50 | 2.16 | 96.76 | 15.91 | 95.33 | 24.33 | 93.49 | 26.62 | 97.26 | 17.04 | 98.29 | 9.75 | 96.77 | 15.97 |
| KL-matching | 97.66 | 9.24 | 94.75 | 22.21 | 93.18 | 28.02 | 88.04 | 40.26 | 85.27 | 55.47 | 91.73 | 33.41 | 91.77 | 31.43 |
| X-Maha (ours) | 100.0 | 0.00 | 99.36 | 3.16 | 94.09 | 31.35 | 99.47 | 2.69 | 99.99 | 0.01 | 99.99 | 0.01 | 98.82 | 6.20 |

Table 12: OOD detection performance on CIFAR-100-LT (ID) on smaller transformers.

| Method | Texture | | SVHN | | CIFAR10 | | Tiny ImageNet | | LSUN | | Places365 | | Average | |
|---|---|---|---|---|---|---|---|---|---|---|---|---|---|---|
| | AUROC | FPR95 | AUROC | FPR95 | AUROC | FPR95 | AUROC | FPR95 | AUROC | FPR95 | AUROC | FPR95 | AUROC | FPR95 |
| VIT_TINY_PATCH16_224 | | | | | | | | | | | | | | |
| MSP | 90.08 | 43.37 | 81.70 | 67.65 | 79.48 | 71.62 | 75.96 | 75.24 | 71.36 | 84.26 | 79.41 | 71.40 | 79.66 | 68.92 |
| MLS | 99.12 | 3.60 | 93.33 | 33.93 | 79.87 | 74.86 | 85.81 | 56.68 | 93.82 | 33.03 | 96.58 | 18.97 | 91.42 | 36.85 |
| Energy | 99.38 | 2.13 | 93.83 | 31.25 | 78.33 | 78.86 | 86.24 | 56.10 | 95.32 | 24.89 | 97.50 | 13.26 | 91.77 | 34.42 |
| Mahalanobis | 99.85 | 0.53 | 97.29 | 12.80 | 85.08 | 63.78 | 91.26 | 35.45 | 98.44 | 8.10 | 98.67 | 6.67 | 95.10 | 21.22 |
| Residual | 99.36 | 2.70 | 85.09 | 63.99 | 63.92 | 86.08 | 86.92 | 56.50 | 95.55 | 23.68 | 95.84 | 23.68 | 88.68 | 42.72 |
| Vim | 99.48 | 1.86 | 94.02 | 30.06 | 78.59 | 78.29 | 86.75 | 53.95 | 95.67 | 22.96 | 97.70 | 12.17 | 92.03 | 33.21 |
| NECO | 99.43 | 2.16 | 93.71 | 31.78 | 80.42 | 73.09 | 87.26 | 50.52 | 95.19 | 24.22 | 97.42 | 14.01 | 92.24 | 32.63 |
| KL-matching | 94.50 | 23.48 | 86.54 | 55.21 | 82.74 | 62.81 | 79.16 | 66.66 | 74.56 | 80.30 | 83.53 | 60.59 | 83.51 | 58.18 |
| X-Maha (ours) | 99.99 | 0.04 | 97.77 | 10.81 | 83.67 | 66.99 | 96.31 | 16.67 | 99.97 | 0.01 | 99.95 | 0.14 | 96.28 | 15.78 |
| VIT_SMALL_PATCH16_224 | | | | | | | | | | | | | | |
| MSP | 96.39 | 16.72 | 92.72 | 37.39 | 87.58 | 49.60 | 82.39 | 57.10 | 80.54 | 68.09 | 87.52 | 49.64 | 87.85 | 46.42 |
| MLS | 99.69 | 1.44 | 95.97 | 21.84 | 91.96 | 41.67 | 92.62 | 29.45 | 97.66 | 14.61 | 98.77 | 6.74 | 96.11 | 19.29 |
| Energy | 99.80 | 1.13 | 95.29 | 27.28 | 91.55 | 45.20 | 93.37 | 25.75 | 98.58 | 8.86 | 99.28 | 3.78 | 96.31 | 18.67 |
| Mahalanobis | 99.91 | 0.53 | 99.43 | 2.35 | 93.02 | 35.98 | 97.15 | 12.93 | 99.65 | 1.67 | 99.59 | 2.35 | 98.12 | 9.30 |
| Residual | 99.93 | 0.25 | 96.22 | 24.49 | 83.28 | 64.13 | 95.96 | 21.20 | 99.26 | 3.78 | 99.37 | 2.78 | 95.67 | 19.44 |
| Vim | 99.84 | 0.96 | 95.74 | 25.12 | 91.69 | 44.68 | 93.78 | 24.60 | 98.77 | 7.65 | 99.38 | 3.29 | 96.53 | 17.72 |
| NECO | 99.77 | 1.13 | 96.30 | 20.92 | 91.64 | 41.49 | 92.86 | 27.03 | 97.67 | 14.01 | 98.92 | 5.90 | 96.19 | 18.41 |
| KL-matching | 98.12 | 7.73 | 95.63 | 20.27 | 90.16 | 37.62 | 85.25 | 46.40 | 83.94 | 56.00 | 90.77 | 36.06 | 90.65 | 34.01 |
| X-Maha (ours) | 100.0 | 0.00 | 99.68 | 1.05 | 92.24 | 39.11 | 99.48 | 2.44 | 100.0 | 0.00 | 100.0 | 0.00 | 98.57 | 7.10 |

## A.5 Additional time consumption analysis

Unlike the direct Mahalanobis distance, which considers only the final layer of features, our approach necessitates the integration of features across all layers. This inevitably leads to additional time consumption. Table 15 presents the time consumption at different stages of the test phase, measured in seconds, on the ImageNet-LT dataset (ID) and the fine-tuned ViT model. "Pre-process" represents the process of pre-processing the ID training set, including the calculation of the mean and covariance matrix required for Mahalanobis distance, with additional importance weights $\alpha$ for X-Maha. Each subsequent column represents the time required to process each dataset including the ID test set and OOD datasets, and the last column represents the total time consumed. From the results, we observe that our approach only brings about an additional $10\%$ total time consumption, but results in an improvement of AUROC by $2.39\%$ and a reduction of FPR95 by $7.66\%$ on average, demonstrating the efficacy of our approach.

## A.6 Fair comparison with MCM

The MCM method is naturally better suited for zero-shot OOD tasks compared to fine-tuning tasks. The prevalent fine-tuning approach, which mainly targets the visual encoder, tends to disrupt the initial alignment between the visual and text components after fine-tuning, resulting in less effective outcomes. Our goal in including the MCM method in our experiment was not to make a

Table 13: OOD detection performance on ImageNet-LT (ID) on smaller transformers.

| Method | Texture AUROC | FPR95 | Places AUROC | FPR95 | SUN AUROC | FPR95 | iNaturalist AUROC | FPR95 | ImageNet-O AUROC | FPR95 | Average AUROC | FPR95 |
|---|---|---|---|---|---|---|---|---|---|---|---|---|
| | | | | | VIT_TINY_PATCH16_224 | | | | | | | |
| MSP | 78.01 | 73.35 | 75.50 | 78.45 | 75.30 | 79.07 | 87.21 | 54.52 | 67.95 | 87.70 | 76.80 | 74.62 |
| MLS | 84.44 | 65.73 | 78.50 | 75.84 | 79.44 | 75.44 | 91.83 | 46.49 | 76.83 | 84.55 | 82.21 | 69.61 |
| Energy | 85.85 | 60.04 | 78.76 | 74.90 | 80.02 | 74.11 | 92.72 | 42.10 | 78.73 | 81.55 | 83.22 | 66.54 |
| Mahalanobis | 89.61 | 41.86 | 79.27 | 67.26 | 82.44 | 63.88 | 97.62 | 11.83 | 80.09 | 77.30 | 85.81 | 52.43 |
| Residual | 84.86 | 56.21 | 68.21 | 86.03 | 69.96 | 84.55 | 88.63 | 49.79 | 73.94 | 77.95 | 77.12 | 70.91 |
| Vim | 86.49 | 57.06 | 78.97 | 74.35 | 80.27 | 73.02 | 93.25 | 38.66 | 79.22 | 80.85 | 83.64 | 64.79 |
| NECO | 86.84 | 56.44 | 79.03 | 73.47 | 80.61 | 72.26 | 94.78 | 30.01 | 79.54 | 79.75 | 84.16 | 52.39 |
| KL-matching | 81.97 | 67.22 | 77.80 | 74.27 | 78.06 | 73.72 | 91.59 | 41.18 | 72.78 | 84.35 | 80.44 | 68.15 |
| X-Maha (ours) | **92.21** | **29.84** | 78.40 | 68.87 | 81.25 | 66.86 | **97.72** | **11.30** | 82.43 | 69.90 | **86.40** | **49.35** |
| | | | | | VIT_SMALL_PATCH16_224 | | | | | | | |
| MSP | 82.60 | 60.11 | 81.41 | 66.59 | 81.97 | 63.84 | 94.31 | 25.69 | 77.74 | 73.60 | 83.61 | 57.97 |
| MLS | 87.94 | 50.51 | 84.97 | 60.73 | 86.46 | 56.44 | 96.63 | 17.04 | 84.64 | 65.40 | 88.13 | 50.02 |
| Energy | 88.96 | 46.03 | 85.45 | 58.10 | 87.20 | 53.27 | 97.09 | 14.06 | 85.91 | 60.85 | 88.92 | 46.46 |
| Mahalanobis | 91.13 | 36.06 | 86.30 | 54.87 | 89.54 | 46.81 | 99.03 | 4.49 | 87.74 | 55.45 | 90.75 | 39.54 |
| Residual | 88.66 | 45.11 | 79.43 | 70.30 | 84.29 | 60.89 | 96.18 | 20.30 | 82.07 | 65.85 | 86.12 | 52.49 |
| Vim | 89.38 | 44.08 | 85.72 | 56.97 | 87.57 | 52.06 | 97.39 | 12.38 | 86.29 | 59.25 | 89.27 | 44.95 |
| NECO | 89.72 | 43.40 | 85.86 | 56.63 | 88.18 | 51.50 | 98.07 | 9.36 | 87.00 | 58.40 | 89.77 | 43.86 |
| KL-matching | 86.01 | 50.67 | 83.63 | 60.68 | 84.80 | 56.71 | 96.68 | 14.62 | 81.90 | 65.35 | 86.60 | 49.61 |
| X-Maha (ours) | **93.35** | **25.94** | 86.18 | 55.21 | 89.52 | **46.74** | **99.13** | **3.99** | 89.13 | 50.20 | **91.46** | **36.42** |

Table 14: OOD detection performance on ImageNet-LT (ID) and five OOD datasets using the zero-shot models without fine-tuning.

| Method | Texture AUROC | FPR95 | Places AUROC | FPR95 | SUN AUROC | FPR95 | iNaturalist AUROC | FPR95 | ImageNet-O AUROC | FPR95 | Average AUROC | FPR95 |
|---|---|---|---|---|---|---|---|---|---|---|---|---|
| | | | | | IMAGENET-21K PRE-TRAINED VIT-SMALL | | | | | | | |
| MSP | 82.24 | 58.21 | 77.75 | 75.34 | 78.32 | 74.35 | 92.41 | 37.39 | 74.87 | 75.70 | 81.12 | 64.20 |
| MLS | 91.03 | 38.14 | 82.05 | 66.33 | 85.45 | 60.21 | 96.08 | 24.66 | 85.79 | 66.60 | 88.08 | 51.19 |
| Energy | 91.76 | 33.60 | **82.07** | **64.62** | 85.92 | 56.38 | 96.00 | 25.85 | 86.59 | 63.60 | 88.47 | 48.81 |
| Mahalanobis | 92.94 | 26.74 | 80.88 | 68.96 | 85.09 | 60.41 | **98.59** | **6.10** | 88.12 | 53.75 | 89.12 | 43.19 |
| Residual | 89.39 | 43.03 | 61.84 | 89.78 | 69.77 | 84.65 | 92.23 | 39.67 | 80.85 | 67.70 | 78.82 | 64.97 |
| Vim | 92.03 | 32.50 | 81.99 | 64.63 | 85.96 | 56.15 | 96.28 | 23.85 | 86.91 | 62.05 | 88.63 | 47.84 |
| NECO | 92.33 | 31.10 | 78.84 | 69.23 | 83.22 | 62.75 | 97.91 | 9.57 | 87.92 | 56.55 | 88.05 | 45.84 |
| KL-matching | 85.44 | 49.66 | 79.42 | 72.35 | 80.89 | 69.74 | 94.90 | 29.76 | 78.69 | 67.90 | 83.87 | 57.88 |
| X-Maha (ours) | **96.57** | **12.43** | 78.75 | 72.76 | 83.04 | 63.38 | 98.28 | 7.48 | **90.59** | 42.60 | 89.45 | 39.73 |
| | | | | | IMAGENET-21K PRE-TRAINED VIT-TINY | | | | | | | |
| MSP | 76.40 | 73.71 | 70.13 | 86.79 | 68.43 | 88.56 | 82.49 | 70.08 | 64.86 | 87.55 | 72.46 | 81.34 |
| MLS | 87.68 | 51.84 | **73.55** | 81.88 | 76.13 | 82.79 | 90.23 | 57.60 | 77.84 | 83.25 | 81.09 | 71.47 |
| Energy | 88.63 | 46.47 | 73.36 | 81.29 | 76.59 | 81.44 | 90.41 | 56.67 | 78.99 | 81.80 | 81.59 | 69.53 |
| Mahalanobis | 91.78 | 30.85 | 73.34 | **79.18** | **77.17** | **76.42** | 94.89 | 28.37 | 80.38 | 75.75 | 83.51 | 58.11 |
| Residual | 88.96 | 43.39 | 53.28 | 95.00 | 56.74 | 94.70 | 74.95 | 79.52 | 74.57 | 78.40 | 69.70 | 78.20 |
| Vim | 89.06 | 44.47 | 73.18 | 81.50 | 76.48 | 81.48 | 90.55 | 56.22 | 79.33 | 81.10 | 81.72 | 68.95 |
| NECO | 90.04 | 41.58 | 72.85 | 81.42 | 76.17 | 81.01 | 94.23 | 36.09 | 80.81 | 77.30 | 82.82 | 63.48 |
| KL-matching | 80.83 | 65.53 | 71.26 | 86.35 | 70.28 | 88.57 | 86.62 | 65.47 | 69.79 | 83.85 | 75.75 | 77.95 |
| X-Maha (ours) | **95.47** | **17.93** | 71.78 | 79.73 | 74.44 | 78.06 | **96.08** | **20.33** | 85.39 | 57.30 | 84.63 | 50.67 |

Table 15: Time consumption (in seconds) comparison between Mahalanobis and X-Maha.

| Dataset | Pre-process | ID test set | Texture | Places | SUN | iNaturalist | ImageNet-O | Total |
|---|---|---|---|---|---|---|---|---|
| Mahalanobis | 685 | 238 | 36 | 61 | 56 | 59 | 14 | 1149 |
| X-Maha | 748 | 291 | 38 | 62 | 60 | 61 | 15 | 1275 |

direct comparison but to empirically showcase that our proposed method enhances OOD detection performance. Conversely, methods like ViM and NECO are methodologically and conceptually more similar to our approach and, therefore, require a more thorough comparison. Moreover, we present the results of MCM on the fine-tuned model (i.e., MCM-tuned) in Table 16 for comparison.

## A.7 Ablation studies on OpenOOD v1.5 benchmark

We conducted our experiment again using the Openood v1.5 [59] benchmark and chose Imagenet-1K-LT as the ID dataset, as shown in Table 17. From our experience, this approach is comparable to

Table 16: Fair comparison with MCM on CIFAR-100, CIFAR-100-LT, and ImageNet-LT ID datasets.

| Method | Texture | | SVHN | | CIFAR10 | | Tiny ImageNet | | LSUN | | Places365 | | Average | |
|---|---|---|---|---|---|---|---|---|---|---|---|---|---|---|
| | AUROC | FPR95 | AUROC | FPR95 | AUROC | FPR95 | AUROC | FPR95 | AUROC | FPR95 | AUROC | FPR95 | AUROC | FPR95 |
| **CIFAR-100** | | | | | | | | | | | | | | |
| MCM-untuned | 72.98 | 92.09 | 90.75 | 63.39 | 75.53 | 88.66 | 65.54 | 93.36 | 50.79 | 99.11 | 60.97 | 97.79 | 69.43 | 89.06 |
| MCM-tuned | 75.33 | 91.38 | 91.55 | 60.96 | 75.60 | 91.03 | 64.07 | 95.40 | 55.14 | 98.93 | 63.71 | 97.67 | 70.90 | 89.23 |
| X-Maha (ours) | 99.90 | 0.02 | 97.41 | 16.80 | 85.35 | 59.44 | 94.85 | 27.10 | 99.89 | 0.10 | 99.91 | 0.03 | 96.22 | 17.25 |
| **CIFAR-100-LT** | | | | | | | | | | | | | | |
| MCM-untuned | 72.98 | 92.09 | 90.75 | 63.39 | 75.53 | 88.66 | 65.54 | 93.36 | 50.79 | 99.11 | 60.97 | 97.79 | 69.43 | 89.06 |
| MCM-tuned | 75.33 | 91.38 | 91.55 | 60.96 | 75.60 | 91.03 | 64.07 | 95.40 | 55.14 | 98.93 | 63.71 | 97.67 | 70.90 | 89.23 |
| X-Maha (ours) | 99.94 | 0.00 | 98.13 | 8.99 | 88.74 | 52.46 | 97.31 | 13.23 | 99.93 | 0.05 | 99.95 | 0.04 | 97.33 | 12.46 |

| Method | Texture | | Places | | SUN | | iNaturalist | | ImageNet-O | | Average | |
|---|---|---|---|---|---|---|---|---|---|---|---|---|
| | AUROC | FPR95 | AUROC | FPR95 | AUROC | FPR95 | AUROC | FPR95 | AUROC | FPR95 | AUROC | FPR95 |
| **IMAGENET-1K-LT** | | | | | | | | | | | | |
| MCM-untuned | 86.11 | 57.77 | 89.77 | 44.69 | 92.57 | 37.59 | 94.61 | 30.91 | 79.51 | 75.70 | 88.51 | 49.33 |
| MCM-tuned | 85.64 | 60.11 | 89.82 | 44.32 | 92.92 | 36.25 | 94.26 | 32.01 | 79.26 | 76.10 | 88.38 | 49.76 |
| X-Maha (ours) | 89.94 | 46.79 | 90.47 | 42.51 | 92.71 | 37.95 | 94.79 | 27.57 | 82.94 | 69.10 | 90.17 | 44.78 |

using ImageNet-1k while being more time-efficient. Our results surpassed those of all other methods by a significant margin on average, highlighting the success of our X-Maha strategy.

Table 17: OOD detection performance on ImageNet-LT (ID) on OpenOOD v1.5.

| Method | NINCO | | Openimage-O | | SSB-Hard | | iImageNet-C | | ImageNet-ES | | iImageNet-R | | ImageNet-V2 | | Average | |
|---|---|---|---|---|---|---|---|---|---|---|---|---|---|---|---|---|
| | AUROC | FPR95 | AUROC | FPR95 | AUROC | FPR95 | AUROC | FPR95 | AUROC | FPR95 | AUROC | FPR95 | AUROC | FPR95 | AUROC | FPR95 |
| **IMAGENET-21K PRE-TRAINED VIT** | | | | | | | | | | | | | | | | |
| MSP | 87.81 | 50.02 | 93.72 | 27.51 | 76.72 | 68.43 | 67.91 | 78.58 | 69.35 | 69.26 | 79.73 | 59.15 | 57.57 | 89.92 | 76.12 | 63.27 |
| MLS | 91.59 | 42.80 | 96.28 | 18.65 | 81.25 | 63.86 | 70.54 | 76.77 | 72.11 | 66.79 | 83.65 | 53.22 | 57.86 | 90.09 | 79.04 | 58.88 |
| Energy | 92.12 | 39.62 | 96.80 | 16.02 | 81.87 | 61.53 | 70.80 | 76.02 | 72.44 | 66.06 | 84.25 | 50.52 | 57.79 | 90.19 | 79.44 | 57.14 |
| Mahalanobis | 94.00 | 32.51 | 97.58 | 12.61 | 85.01 | 52.17 | 73.93 | 72.64 | 73.04 | 67.08 | 85.32 | 48.95 | 58.02 | 90.81 | 80.99 | 53.83 |
| Residual | 83.87 | 62.45 | 92.41 | 33.88 | 84.87 | 56.19 | 74.96 | 78.03 | 65.25 | 82.87 | 75.05 | 76.46 | 53.03 | 94.38 | 75.63 | 69.18 |
| Vim | 92.29 | 38.65 | 96.94 | 15.32 | 82.36 | 60.40 | 71.19 | 75.39 | 72.47 | 65.98 | 84.37 | 50.20 | 57.79 | 90.15 | 79.63 | 56.58 |
| NECO | 91.97 | 38.09 | 96.90 | 15.19 | 84.81 | 54.96 | 70.55 | 75.48 | 72.01 | 67.61 | 82.43 | 53.86 | 56.86 | 90.44 | 79.36 | 56.52 |
| KL-matching | 90.53 | 41.63 | 95.95 | 18.15 | 79.52 | 63.02 | 70.03 | 75.91 | 71.54 | 66.35 | 82.56 | 52.60 | 58.33 | 89.85 | 78.35 | 58.22 |
| X-Maha (ours) | 94.98 | 26.74 | 98.21 | 9.72 | 86.34 | 49.46 | 83.96 | 53.57 | 76.78 | 63.45 | 88.49 | 42.85 | 58.36 | 91.36 | 83.88 | 48.16 |
| **CLIP-VIT-B/16** | | | | | | | | | | | | | | | | |
| MSP | 80.11 | 68.94 | 88.22 | 46.72 | 68.06 | 83.66 | 73.44 | 70.44 | 70.31 | 68.46 | 77.27 | 64.00 | 57.12 | 90.78 | 73.50 | 70.43 |
| MLS | 84.17 | 67.11 | 92.93 | 35.17 | 71.99 | 82.44 | 77.66 | 67.65 | 75.66 | 64.21 | 84.61 | 55.96 | 58.24 | 90.38 | 77.89 | 66.13 |
| Energy | 84.15 | 67.78 | 93.73 | 30.59 | 72.25 | 82.43 | 78.02 | 67.13 | 76.51 | 62.51 | 85.87 | 52.26 | 58.22 | 90.28 | 78.39 | 64.71 |
| Mahalanobis | 75.13 | 83.28 | 86.95 | 63.82 | 66.11 | 89.49 | 82.68 | 62.64 | 84.27 | 52.46 | 90.02 | 47.33 | 58.18 | 90.31 | 77.62 | 69.90 |
| Residual | 61.56 | 91.54 | 70.43 | 81.35 | 61.00 | 92.63 | 82.86 | 68.28 | 86.30 | 54.65 | 86.28 | 57.41 | 56.47 | 92.06 | 72.13 | 76.85 |
| Vim | 83.91 | 68.20 | 93.57 | 31.15 | 72.28 | 82.57 | 78.85 | 65.06 | 77.52 | 60.34 | 86.73 | 49.79 | 58.35 | 89.88 | 78.74 | 63.86 |
| NECO | 880.96 | 71.90 | 92.48 | 37.13 | 69.22 | 85.10 | 77.84 | 68.17 | 78.69 | 61.76 | 86.00 | 54.92 | 58.18 | 89.83 | 77.63 | 66.97 |
| KL-matching | 83.21 | 66.88 | 92.13 | 34.08 | 70.63 | 81.75 | 76.09 | 67.07 | 72.92 | 64.25 | 81.70 | 55.46 | 57.83 | 90.51 | 76.36 | 65.71 |
| X-Maha (ours) | 8.30 | 74.67 | 92.90 | 36.93 | 71.64 | 81.79 | 85.49 | 50.94 | 87.84 | 43.49 | 88.76 | 50.31 | 58.22 | 89.50 | 80.45 | 61.09 |

## A.8 Experiments on OpenOOD v1.5 benchmark

Table 18 presents a comprehensive evaluation of the EVA model's out-of-distribution detection performance using the ImageNet-LT dataset under the OpenOOD v1.5 evaluation framework.

Table 18: OOD detection performance on ImageNet-LT (ID) on OpenOOD v1.5 on EVA.

| Method | NINCO | | Openimage-O | | SSB-Hard | | iImageNet-C | | ImageNet-ES | | iImageNet-R | | ImageNet-V2 | | Average | |
|---|---|---|---|---|---|---|---|---|---|---|---|---|---|---|---|---|
| | AUROC | FPR95 | AUROC | FPR95 | AUROC | FPR95 | AUROC | FPR95 | AUROC | FPR95 | AUROC | FPR95 | AUROC | FPR95 | AUROC | FPR95 |
| **EVA02-SMALL-PATCH14-336** | | | | | | | | | | | | | | | | |
| MSP | 85.42 | 58.74 | 93.96 | 28.32 | 67.21 | 81.62 | 71.52 | 70.71 | 71.12 | 65.98 | 82.42 | 50.02 | 56.44 | 90.10 | 75.44 | 63.64 |
| MLS | 85.22 | 59.42 | 93.98 | 28.32 | 66.81 | 81.78 | 70.94 | 71.24 | 72.48 | 65.16 | 82.91 | 49.36 | 56.44 | 90.19 | 75.54 | 63.64 |
| Energy | 60.93 | 89.79 | 70.32 | 86.76 | 51.36 | 95.11 | 50.30 | 95.21 | 70.05 | 74.32 | 71.41 | 79.83 | 51.72 | 93.81 | 60.87 | 87.83 |
| Mahalanobis | 88.50 | 52.47 | 95.35 | 24.99 | 73.67 | 73.68 | 74.99 | 66.73 | 74.47 | 64.15 | 86.59 | 47.58 | 57.97 | 90.15 | 78.75 | 59.97 |
| Residual | 46.13 | 99.22 | 61.80 | 96.84 | 48.91 | 95.38 | 59.63 | 92.57 | 57.34 | 91.72 | 64.01 | 89.43 | 48.01 | 96.41 | 55.12 | 94.51 |
| Vim | 46.94 | 99.00 | 63.01 | 96.48 | 49.00 | 95.31 | 59.52 | 92.49 | 58.76 | 91.23 | 65.12 | 88.46 | 48.13 | 96.33 | 54.52 | 94.87 |
| NECO | 79.29 | 63.75 | 93.32 | 28.33 | 61.44 | 83.42 | 68.80 | 72.11 | 70.25 | 66.85 | 81.47 | 49.12 | 54.27 | 91.34 | 72.69 | 64.29 |
| KNN | 85.02 | 68.02 | 93.33 | 39.61 | 68.99 | 86.43 | 74.23 | 68.85 | 75.50 | 63.43 | 86.55 | 45.53 | 57.85 | 90.52 | 77.35 | 66.06 |
| NNguide | 84.97 | 68.02 | 93.33 | 39.47 | 68.90 | 86.47 | 74.11 | 68.92 | 75.67 | 63.36 | 86.58 | 45.44 | 57.83 | 90.52 | 77.34 | 66.03 |
| RelativeMaha | 89.44 | 50.54 | 95.04 | 25.24 | 74.26 | 73.55 | 73.82 | 68.31 | 73.32 | 64.74 | 85.49 | 48.87 | 58.37 | 89.66 | 78.53 | 60.13 |
| KL-matching | 12.43 | 99.97 | 7.23 | 99.99 | 27.79 | 98.64 | 27.30 | 99.01 | 26.64 | 98.68 | 13.68 | 99.77 | 42.69 | 96.04 | 22.54 | 98.87 |
| X-Maha | 88.71 | 51.72 | 95.50 | 24.17 | 74.39 | 72.80 | 76.70 | 65.17 | 74.30 | 64.14 | 87.15 | 46.55 | 58.12 | 90.09 | 79.27 | 59.23 |

## A.9 Ablation studies on varying parameter-efficient fine-tuning methods.

X-Maha is a general framework in which many lightweight fine-tuning methods can be integrated. In addition to Adaptformer [4] which is used in our experiments by default, we test X-Maha with

another 5 parameter-efficient fine-tuning (PEFT) methods as well as full fine-tuning. Specifically, we combine X-Maha with *Bias-tuning* [58], *VPT-shallow* [21], *VPT-deep* [21], *LoRA* [18], and *Adapter* [16]. We report the empirical results for CIFAR-100 in Table 19, CIFAR-100-LT in Table 20, and ImageNet-LT in Table 21. From the results, we observe that X-Maha consistently improves the baselines by a large margin, showing its robustness to the PEFT methods.

Table 19: OOD detection performance in terms of AUROC (↑) and FPR95 (↓) for different PEFT methods, and full fine-tuning on CIFAR-100 dataset.

| Method | Texture | | SVHN | | CIFAR10 | | Tiny ImageNet | | LSUN | | Places | | **Average** | |
|---|---|---|---|---|---|---|---|---|---|---|---|---|---|---|
| | AUROC | FPR95 | AUROC | FPR95 | AUROC | FPR95 | AUROC | FPR95 | AUROC | FPR95 | AUROC | FPR95 | AUROC | FPR95 |
| **Bias-tuning** | | | | | | | | | | | | | | |
| + MSP | 97.46 | 12.77 | 94.72 | 27.13 | 94.19 | 29.44 | 88.42 | 44.58 | 86.25 | 64.76 | 91.93 | 41.80 | 92.16 | 36.75 |
| + MLS | 99.71 | 1.24 | 96.60 | 13.31 | 96.96 | 14.23 | 94.69 | 22.00 | 98.07 | 10.86 | 98.73 | 6.84 | 97.46 | 11.41 |
| + Energy | 99.82 | 0.87 | 96.62 | 12.60 | 97.07 | 13.58 | 95.10 | 20.07 | 98.72 | 6.86 | 99.11 | 4.74 | 97.74 | 9.79 |
| + Mahalanobis | 99.93 | 0.34 | 98.80 | 5.32 | 96.18 | 20.60 | 97.31 | 10.71 | 99.58 | 1.32 | 99.45 | 2.68 | 98.54 | 6.83 |
| + Residual | 99.98 | 0.04 | 97.20 | 16.92 | 90.96 | 49.13 | 98.46 | 6.69 | 99.90 | 0.14 | 99.81 | 0.59 | 97.72 | 12.25 |
| + Vim | 99.85 | 0.74 | 96.84 | 11.84 | 97.09 | 13.58 | 95.48 | 18.94 | 98.96 | 5.21 | 99.24 | 3.90 | 97.91 | 9.04 |
| + NECO | 99.77 | 1.21 | 97.03 | 12.42 | 96.95 | 15.77 | 94.97 | 20.55 | 98.34 | 10.19 | 98.84 | 6.58 | 97.65 | 11.12 |
| + X-Maha (ours) | 100.0 | 0.00 | 99.44 | 2.29 | 95.32 | 25.35 | 99.68 | 1.52 | 99.99 | 0.01 | 99.99 | 0.02 | **99.07** | **4.87** |
| **VPT-shallow** | | | | | | | | | | | | | | |
| + MSP | 95.84 | 18.09 | 93.78 | 34.50 | 92.09 | 37.35 | 85.90 | 49.15 | 79.15 | 78.05 | 87.17 | 54.51 | 88.99 | 45.27 |
| + MLS | 98.77 | 5.28 | 96.55 | 18.36 | 94.42 | 25.10 | 86.29 | 47.31 | 88.68 | 59.55 | 92.94 | 35.22 | 92.24 | 31.81 |
| + Energy | 99.04 | 4.57 | 96.58 | 15.87 | 94.42 | 24.75 | 85.83 | 51.00 | 89.64 | 55.53 | 93.47 | 32.64 | 93.16 | 30.73 |
| + Mahalanobis | 99.97 | 0.18 | 92.41 | 44.63 | 93.84 | 32.15 | 98.04 | 9.23 | 99.86 | 0.18 | 99.77 | 0.88 | 97.31 | 14.54 |
| + Residual | 99.98 | 0.05 | 80.46 | 67.16 | 86.92 | 55.54 | 99.02 | 5.17 | 99.95 | 0.10 | 99.89 | 0.37 | 94.37 | 21.40 |
| + Vim | 99.29 | 3.62 | 96.71 | 15.16 | 94.57 | 24.64 | 87.34 | 45.83 | 91.64 | 46.32 | 94.70 | 26.97 | 94.04 | 27.09 |
| + NECO | 99.30 | 3.56 | 95.99 | 25.70 | 95.02 | 24.55 | 90.59 | 34.88 | 94.24 | 34.05 | 96.17 | 20.60 | 95.22 | 23.89 |
| + X-Maha (ours) | 100.0 | 0.00 | 94.28 | 36.41 | 92.37 | 38.48 | 99.78 | 1.15 | 99.99 | 0.01 | 99.98 | 0.06 | **97.73** | **12.68** |
| **VPT-deep** | | | | | | | | | | | | | | |
| + MSP | 97.43 | 13.49 | 91.72 | 44.53 | 94.33 | 30.07 | 86.93 | 48.16 | 84.23 | 69.02 | 91.10 | 47.98 | 90.79 | 42.21 |
| + MLS | 99.69 | 12.49 | 96.55 | 15.98 | 96.53 | 30.65 | 95.81 | 25.68 | 97.51 | 13.26 | 97.51 | 13.26 | 96.34 | 16.59 |
| + Energy | 99.79 | 1.12 | 97.59 | 10.49 | 96.53 | 15.82 | 91.43 | 29.60 | 96.53 | 21.24 | 97.95 | 11.21 | 96.64 | 14.91 |
| + Mahalanobis | 99.94 | 0.30 | 94.27 | 39.67 | 96.08 | 22.59 | 97.10 | 162.99 | 99.08 | 4.88 | 99.16 | 4.47 | 97.60 | 14.15 |
| + Residual | 99.97 | 0.04 | 91.25 | 53.35 | 89.88 | 50.67 | 98.07 | 10.16 | 99.69 | 0.74 | 99.58 | 1.64 | 96.41 | 19.43 |
| + Vim | 99.83 | 0.83 | 97.68 | 10.13 | 96.57 | 15.91 | 92.09 | 27.75 | 97.05 | 18.26 | 98.22 | 9.96 | 96.91 | 13.81 |
| + NECO | 99.72 | 1.44 | 96.53 | 17.02 | 96.71 | 17.06 | 92.73 | 26.10 | 96.80 | 20.68 | 98.03 | 11.12 | 96.75 | 15.57 |
| + X-Maha (ours) | 99.99 | 0.02 | 96.36 | 25.31 | 95.33 | 26.39 | 99.59 | 2.03 | 99.95 | 0.00 | 99.93 | 0.18 | **98.52** | **8.99** |
| **LoRA** | | | | | | | | | | | | | | |
| + MSP | 97.36 | 12.77 | 94.85 | 29.23 | 94.36 | 29.49 | 87.26 | 46.89 | 84.76 | 68.83 | 90.95 | 45.35 | 91.59 | 38.76 |
| + MLS | 99.57 | 1.91 | 97.88 | 8.89 | 96.98 | 14.76 | 89.51 | 34.34 | 95.70 | 27.11 | 97.68 | 12.53 | 96.22 | 16.59 |
| + Energy | 99.68 | 1.38 | 98.09 | 7.79 | 97.09 | 14.28 | 89.57 | 34.75 | 96.37 | 23.26 | 98.09 | 10.67 | 96.48 | 15.36 |
| + Mahalanobis | 99.96 | 0.11 | 99.33 | 2.69 | 96.65 | 17.98 | 97.72 | 9.47 | 99.39 | 2.07 | 99.47 | 2.35 | 98.76 | 5.78 |
| + Residual | 99.99 | 0.02 | 98.15 | 9.65 | 91.25 | 44.12 | 98.85 | 4.83 | 99.84 | 0.14 | 99.80 | 0.45 | 97.98 | 9.87 |
| + Vim | 99.75 | 1.13 | 98.29 | 6.91 | 97.121 | 14.25 | 90.50 | 32.38 | 96.96 | 20.00 | 98.38 | 9.27 | 96.83 | 13.99 |
| + NECO | 99.69 | 1.67 | 98.43 | 6.15 | 96.98 | 15.96 | 91.95 | 27.74 | 96.66 | 21.84 | 98.17 | 9.97 | 96.98 | 13.89 |
| + X-Maha (ours) | 100.0 | 0.00 | 99.78 | 0.88 | 95.99 | 21.52 | 99.82 | 0.96 | 99.99 | 0.00 | 99.99 | 0.01 | **99.26** | **3.89** |
| **Adapter** | | | | | | | | | | | | | | |
| + MSP | 97.34 | 12.54 | 95.56 | 23.93 | 91.73 | 38.80 | 85.30 | 48.04 | 84.70 | 62.47 | 90.66 | 42.81 | 90.88 | 38.10 |
| + MLS | 99.90 | 0.32 | 98.31 | 8.01 | 94.26 | 29.06 | 92.18 | 29.58 | 99.10 | 3.69 | 99.50 | 1.51 | 97.21 | 12.03 |
| + Energy | 99.93 | 0.18 | 98.28 | 7.56 | 94.26 | 34.00 | 92.43 | 28.42 | 99.48 | 1.43 | 99.71 | 0.61 | 97.25 | 12.03 |
| + Mahalanobis | 99.97 | 0.12 | 99.44 | 1.82 | 95.04 | 26.43 | 97.58 | 9.60 | 99.55 | 1.76 | 99.63 | 1.36 | 98.53 | 6.85 |
| + Residual | 99.98 | 0.02 | 97.78 | 14.80 | 86.21 | 64.68 | 98.51 | 6.91 | 99.85 | 0.47 | 99.84 | 0.36 | 97.03 | 14.54 |
| + Vim | 99.95 | 0.16 | 98.48 | 6.36 | 93.77 | 33.18 | 93.09 | 26.10 | 99.57 | 0.98 | 99.75 | 0.45 | 97.44 | 11.20 |
| + NECO | 99.90 | 0.30 | 98.60 | 6.92 | 94.41 | 27.76 | 92.31 | 26.44 | 98.91 | 6.70 | 99.45 | 2.68 | 97.26 | 11.82 |
| + X-Maha (ours) | 100.0 | 0.00 | 99.50 | 2.54 | 96.79 | 18.07 | 99.72 | 1.50 | 100.0 | 0.00 | 100.0 | 0.00 | **99.33** | **3.68** |
| **Full fine-tuning** | | | | | | | | | | | | | | |
| + MSP | 97.24 | 15.39 | 91.45 | 46.78 | 93.64 | 33.62 | 87.79 | 48.74 | 85.44 | 72.87 | 91.58 | 48.41 | 91.19 | 44.30 |
| + MLS | 99.72 | 1.12 | 90.65 | 36.84 | 96.55 | 16.03 | 90.43 | 30.61 | 97.84 | 11.13 | 98.97 | 3.63 | 95.69 | 15.56 |
| + Energy | 99.76 | 0.89 | 90.44 | 38.61 | 96.57 | 15.89 | 90.47 | 30.36 | 98.11 | 9.60 | 99.13 | 3.08 | 95.75 | 16.40 |
| + Mahalanobis | 99.87 | 0.55 | 96.80 | 16.06 | 96.87 | 15.38 | 97.46 | 13.26 | 97.69 | 16.25 | 98.96 | 6.61 | 97.94 | 11.35 |
| + Residual | 99.98 | 0.12 | 98.13 | 9.62 | 95.11 | 26.57 | 99.13 | 4.86 | 99.86 | 1.04 | 99.86 | 0.54 | **98.65** | **7.13** |
| + Vim | 99.82 | 0.57 | 91.57 | 34.06 | 96.64 | 15.50 | 91.95 | 25.90 | 98.39 | 7.19 | 99.28 | 2.27 | 96.27 | 14.25 |
| + NECO | 99.71 | 1.33 | 93.01 | 31.88 | 96.96 | 15.53 | 92.41 | 26.21 | 97.35 | 17.86 | 98.78 | 6.72 | 96.37 | 16.59 |
| + X-Maha (ours) | 99.93 | 0.32 | 97.12 | 14.56 | 96.91 | 15.18 | 97.92 | 11.72 | 98.64 | 8.87 | 99.40 | 3.41 | 98.32 | 9.01 |

## Limitations and Broader Impacts

**Limitations** Despite X-Maha's superior performance compared to the existing methods, it exhibits certain limitations, and there are several unexplored research avenues. For example, the current algorithm only provides a simple approach to calculate Transformer feature mixing weights, which might not be optimal. In addition, our method assumes consistent feature dimensions across all layers, which limits the applicability for more neural network architectures.

Table 20: OOD detection performance in terms of AUROC (↑) and FPR95 (↓) for different PEFT methods, and full fine-tuning on CIFAR-100-LT dataset.

| Method | Texture | | SVHN | | CIFAR10 | | Tiny ImageNet | | LSUN | | Places | | Average | |
|---|---|---|---|---|---|---|---|---|---|---|---|---|---|---|
| | AUROC | FPR95 | AUROC | FPR95 | AUROC | FPR95 | AUROC | FPR95 | AUROC | FPR95 | AUROC | FPR95 | AUROC | FPR95 |
| **Bias-tuning** | | | | | | | | | | | | | | |
| + MSP | 97.23 | 12.91 | 95.68 | 23.01 | 91.66 | 38.32 | 85.10 | 49.19 | 83.91 | 65.44 | 89.82 | 45.37 | 90.56 | 39.04 |
| + MLS | 99.89 | 0.37 | 97.73 | 10.36 | 94.29 | 28.15 | 93.67 | 24.36 | 98.87 | 5.06 | 99.29 | 2.91 | 97.29 | 11.87 |
| + Energy | 99.93 | 0.25 | 97.42 | 13.18 | 93.78 | 33.19 | 94.14 | 22.64 | 99.38 | 2.08 | 99.59 | 1.38 | 97.37 | 12.12 |
| + Mahalanobis | 99.96 | 0.20 | 99.58 | 1.27 | 94.59 | 28.56 | 97.26 | 10.39 | 99.54 | 2.14 | 99.55 | 1.84 | 98.40 | 7.40 |
| + Residual | 99.97 | 0.05 | 97.98 | 11.87 | 85.50 | 67.10 | 98.00 | 9.44 | 99.77 | 0.53 | 99.74 | 0.71 | 96.83 | 14.95 |
| + Vim | 99.95 | 0.16 | 97.74 | 10.84 | 93.87 | 32.51 | 94.67 | 20.93 | 99.50 | 1.49 | 99.66 | 0.99 | 97.57 | 11.15 |
| + NECO | 99.89 | 0.51 | 98.15 | 8.82 | 94.32 | 27.12 | 93.33 | 22.52 | 98.59 | 9.34 | 99.23 | 4.02 | 97.25 | 12.06 |
| + X-Maha (ours) | 100.0 | 0.00 | 99.91 | 0.08 | 93.33 | 35.86 | 99.76 | 1.18 | 100.0 | 0.00 | 100.0 | 0.00 | **98.83** | **6.19** |
| **VPT-shallow** | | | | | | | | | | | | | | |
| + MSP | 94.99 | 22.66 | 94.31 | 32.06 | 88.65 | 52.40 | 82.64 | 58.58 | 78.45 | 82.38 | 85.96 | 59.76 | 87.50 | 51.31 |
| + MLS | 99.43 | 2.85 | 96.76 | 18.52 | 87.77 | 54.41 | 81.14 | 64.50 | 93.33 | 40.85 | 95.01 | 27.59 | 92.24 | 34.79 |
| + Energy | 99.61 | 1.72 | 96.05 | 24.39 | 86.05 | 62.46 | 79.32 | 73.40 | 94.82 | 31.44 | 95.72 | 23.16 | 91.93 | 36.09 |
| + Mahalanobis | 99.92 | 0.37 | 93.09 | 38.88 | 91.12 | 42.84 | 96.48 | 14.27 | 99.64 | 1.34 | 99.57 | 1.82 | 96.64 | 16.59 |
| + Residual | 99.92 | 0.28 | 84.78 | 53.96 | 80.83 | 70.84 | 97.42 | 13.56 | 99.75 | 0.76 | 99.61 | 1.35 | 93.72 | 23.46 |
| + Vim | 99.72 | 1.40 | 96.26 | 22.38 | 86.39 | 60.61 | 81.39 | 67.38 | 95.97 | 24.05 | 96.55 | 18.72 | 92.72 | 32.42 |
| + NECO | 99.70 | 1.37 | 96.04 | 24.00 | 89.82 | 43.79 | 86.69 | 43.97 | 95.27 | 24.19 | 97.75 | 16.80 | 94.05 | 25.69 |
| + X-Maha (ours) | 100.0 | 0.02 | 95.65 | 25.96 | 88.82 | 50.14 | 99.72 | 1.36 | 99.98 | 0.00 | 99.97 | 0.04 | **97.36** | **12.92** |
| **VPT-deep** | | | | | | | | | | | | | | |
| + MSP | 96.78 | 14.73 | 92.13 | 38.71 | 90.87 | 42.17 | 83.57 | 53.66 | 81.08 | 72.16 | 87.71 | 52.15 | 88.69 | 45.60 |
| + MLS | 99.78 | 0.87 | 97.63 | 11.72 | 90.42 | 44.26 | 87.38 | 45.51 | 96.43 | 22.13 | 97.87 | 12.18 | 94.92 | 22.78 |
| + Energy | 99.86 | 0.55 | 97.75 | 10.55 | 88.92 | 54.12 | 87.04 | 49.12 | 97.25 | 16.66 | 98.35 | 9.25 | 94.86 | 23.37 |
| + Mahalanobis | 99.88 | 0.39 | 98.22 | 10.49 | 92.63 | 40.93 | 95.85 | 16.72 | 98.81 | 6.74 | 98.94 | 5.10 | 97.39 | 13.39 |
| + Residual | 99.90 | 0.30 | 95.61 | 23.77 | 82.05 | 73.28 | 96.25 | 18.13 | 99.23 | 3.43 | 99.11 | 3.77 | 95.36 | 20.45 |
| + Vim | 99.89 | 0.50 | 98.00 | 9.27 | 89.13 | 53.34 | 88.05 | 45.00 | 97.68 | 14.16 | 98.58 | 7.80 | 95.22 | 21.68 |
| + NECO | 99.82 | 0.78 | 97.78 | 11.74 | 91.39 | 39.08 | 89.00 | 35.93 | 96.60 | 18.40 | 98.14 | 10.10 | 95.45 | 19.34 |
| + X-Maha (ours) | 99.99 | 0.00 | 99.32 | 2.97 | 91.05 | 46.61 | 99.56 | 2.10 | 99.94 | 0.00 | 99.92 | 0.07 | **98.30** | **8.62** |
| **LoRA** | | | | | | | | | | | | | | |
| + MSP | 96.77 | 15.05 | 94.10 | 32.79 | 91.25 | 41.07 | 84.06 | 51.62 | 81.70 | 71.24 | 88.80 | 49.24 | 89.45 | 43.50 |
| + MLS | 99.78 | 0.85 | 96.95 | 16.78 | 93.06 | 32.74 | 87.54 | 43.19 | 97.66 | 14.45 | 98.75 | 6.61 | 95.62 | 19.10 |
| + Energy | 99.84 | 0.44 | 96.51 | 20.22 | 92.28 | 39.13 | 87.19 | 47.63 | 98.31 | 9.71 | 99.09 | 4.23 | 95.54 | 20.23 |
| + Mahalanobis | 99.97 | 0.09 | 99.59 | 1.12 | 94.16 | 30.32 | 97.26 | 10.76 | 99.47 | 2.15 | 99.59 | 1.55 | 98.34 | 7.66 |
| + Residual | 99.98 | 0.07 | 98.04 | 11.86 | 84.77 | 66.35 | 98.08 | 8.99 | 99.74 | 0.77 | 99.72 | 0.70 | 96.72 | 14.79 |
| + Vim | 99.89 | 0.39 | 97.00 | 16.42 | 92.41 | 38.74 | 88.48 | 42.91 | 98.63 | 7.75 | 99.25 | 3.28 | 95.95 | 18.25 |
| + NECO | 99.84 | 0.69 | 97.89 | 11.25 | 93.25 | 31.42 | 89.67 | 33.65 | 97.47 | 15.13 | 98.84 | 6.11 | 96.16 | 16.38 |
| + X-Maha (ours) | 100.0 | 0.00 | 99.96 | 0.01 | 92.62 | 38.01 | 99.87 | 0.59 | 100.0 | 0.00 | 100.0 | 0.00 | **98.74** | **6.43** |
| **Adapter** | | | | | | | | | | | | | | |
| + MSP | 97.34 | 12.54 | 95.56 | 23.93 | 91.73 | 38.80 | 85.30 | 48.04 | 84.70 | 62.47 | 90.66 | 42.81 | 90.88 | 38.10 |
| + MLS | 99.90 | 0.32 | 98.31 | 8.01 | 94.26 | 29.06 | 92.18 | 29.58 | 99.10 | 3.69 | 99.50 | 1.51 | 97.21 | 12.03 |
| + Energy | 99.93 | 0.18 | 98.28 | 7.56 | 93.67 | 34.00 | 92.43 | 28.42 | 99.48 | 1.43 | 99.71 | 0.61 | 97.25 | 12.03 |
| + Mahalanobis | 99.97 | 0.12 | 99.44 | 1.82 | 95.04 | 26.43 | 97.58 | 9.60 | 99.55 | 1.76 | 99.63 | 1.36 | 98.53 | 6.85 |
| + Residual | 99.98 | 0.02 | 97.78 | 14.80 | 86.21 | 64.68 | 98.51 | 6.91 | 99.85 | 0.47 | 99.84 | 0.36 | 97.03 | 14.54 |
| + Vim | 99.95 | 0.16 | 98.48 | 6.36 | 93.77 | 33.18 | 93.09 | 26.10 | 99.57 | 0.98 | 99.75 | 0.45 | 97.44 | 11.20 |
| + NECO | 99.90 | 0.30 | 98.60 | 6.92 | 94.41 | 27.76 | 92.31 | 26.44 | 98.91 | 6.70 | 99.45 | 2.68 | 97.26 | 11.82 |
| + X-Maha (ours) | 100.0 | 0.00 | 99.92 | 0.09 | 94.12 | 30.70 | 99.80 | 0.92 | 100.0 | 0.00 | 100.0 | 0.00 | **98.97** | **5.28** |
| **Full fine-tuning** | | | | | | | | | | | | | | |
| + MSP | 96.93 | 14.29 | 93.98 | 32.33 | 90.46 | 46.33 | 83.91 | 53.60 | 85.16 | 66.75 | 90.43 | 46.44 | 90.14 | 43.29 |
| + MLS | 99.86 | 0.50 | 94.01 | 33.58 | 93.96 | 28.83 | 88.43 | 37.80 | 99.16 | 1.93 | 99.41 | 1.80 | 95.81 | 17.41 |
| + Energy | 99.91 | 0.34 | 92.59 | 47.61 | 93.84 | 29.66 | 88.46 | 37.68 | 99.53 | 0.71 | 99.64 | 0.97 | 95.66 | 19.49 |
| + Mahalanobis | 99.95 | 0.27 | 97.08 | 16.06 | 95.14 | 23.82 | 97.04 | 13.62 | 99.18 | 5.22 | 99.52 | 2.62 | 97.99 | 10.27 |
| + Residual | 99.99 | 0.02 | 97.71 | 13.20 | 90.08 | 49.81 | 98.89 | 5.16 | 99.86 | 0.35 | 99.91 | 0.26 | 97.74 | 11.47 |
| + Vim | 99.94 | 0.25 | 93.63 | 40.38 | 93.98 | 29.20 | 90.18 | 32.65 | 99.65 | 0.41 | 99.73 | 0.65 | 96.18 | 17.26 |
| + NECO | 99.87 | 0.57 | 94.73 | 29.96 | 94.34 | 25.66 | 90.05 | 31.81 | 98.75 | 8.39 | 99.27 | 3.89 | 96.17 | 16.71 |
| + X-Maha (ours) | 99.99 | 0.02 | 97.86 | 10.72 | 95.20 | 23.66 | 98.50 | 38.37 | 99.81 | 80.77 | 99.87 | 0.58 | **98.54** | **7.35** |

**Broader Impacts** This study falls within the domain of out-of-distribution (OOD) detection, a machine learning paradigm that aims to achieve superior classification performance in known classes while identifying OOD samples. Consequently, as this technique gains efficacy and wider adoption, the necessity for extensive data annotation may get diminished, potentially contributing to a rise in unemployment among data annotation professionals.

Table 21: OOD detection performance in terms of AUROC (↑) and FPR95 (↓) for different PEFT methods, and full fine-tuning on ImageNet-LT dataset.

| Method | Texture | | Places | | SUN | | iNaturalist | | ImageNet-O | | Average | |
| | AUROC | FPR95 | AUROC | FPR95 | AUROC | FPR95 | AUROC | FPR95 | AUROC | FPR95 | AUROC | FPR95 |
|---|---|---|---|---|---|---|---|---|---|---|---|---|
| **Bias-tuning** | | | | | | | | | | | | |
| + MSP | 83.92 | 58.30 | 80.95 | 67.86 | 81.16 | 66.33 | 94.31 | 26.99 | 75.83 | 77.60 | 83.23 | 59.42 |
| + MLS | 88.71 | 49.75 | 84.46 | 61.47 | 85.84 | 58.57 | 96.93 | 16.31 | 82.68 | 71.40 | 87.72 | 51.50 |
| + Energy | 89.93 | 43.48 | 85.15 | 57.30 | 86.97 | 52.85 | 97.84 | 10.11 | 84.22 | 66.95 | 88.82 | 46.14 |
| + Mahalanobis | 87.55 | 59.63 | 82.72 | 61.63 | 86.38 | 51.55 | 97.89 | 10.89 | 85.03 | 62.65 | 87.92 | 49.27 |
| + Residual | 73.74 | 81.88 | 68.43 | 82.87 | 75.65 | 73.07 | 88.27 | 48.56 | 72.40 | 80.00 | 75.70 | 73.28 |
| + Vim | 90.06 | 42.68 | 85.22 | 57.05 | 87.16 | 52.08 | 97.92 | 9.83 | 84.39 | 66.60 | **88.95** | **45.65** |
| + NECO | 88.38 | 50.51 | 84.00 | 60.84 | 86.25 | 57.23 | 97.65 | 12.16 | 83.85 | 67.40 | 88.03 | 49.63 |
| + X-Maha (ours) | 90.95 | 41.45 | 81.67 | 63.98 | 85.48 | 54.88 | 97.75 | 11.44 | 86.58 | 56.70 | 88.48 | 45.69 |
| **VPT-shallow** | | | | | | | | | | | | |
| + MSP | 85.58 | 49.11 | 85.44 | 55.99 | 86.38 | 52.79 | 97.53 | 10.26 | 83.79 | 57.95 | 87.74 | 45.22 |
| + MLS | 89.30 | 41.47 | 88.52 | 49.37 | 90.05 | 45.47 | 98.64 | 5.90 | 88.74 | 48.50 | 91.05 | 38.14 |
| + Energy | 90.00 | 37.94 | 89.01 | 46.13 | 90.73 | 41.65 | 98.95 | 4.31 | 89.69 | 44.20 | 91.68 | 34.85 |
| + Mahalanobis | 92.07 | 29.52 | 86.20 | 58.31 | 88.98 | 49.94 | 99.19 | 3.10 | 91.51 | 38.95 | 91.59 | 35.96 |
| + Residual | 88.37 | 49.73 | 73.68 | 80.55 | 79.27 | 72.40 | 96.66 | 16.66 | 84.03 | 60.05 | 84.40 | 55.88 |
| + Vim | 90.32 | 36.33 | 89.04 | 46.03 | 90.82 | 41.24 | 99.02 | 4.11 | 89.93 | 42.75 | 91.83 | 34.09 |
| + NECO | 91.15 | 33.32 | 87.20 | 49.53 | 89.67 | 44.33 | 99.01 | 3.86 | 89.98 | 43.80 | 91.40 | 34.97 |
| + X-Maha (ours) | 95.93 | 14.54 | 85.98 | 57.66 | 89.03 | 47.59 | 99.18 | 3.17 | 93.34 | 32.30 | **92.69** | **31.05** |
| **VPT-deep** | | | | | | | | | | | | |
| + MSP | 85.28 | 49.27 | 84.75 | 57.12 | 85.92 | 53.82 | 97.13 | 11.51 | 83.13 | 58.20 | 87.24 | 45.98 |
| + MLS | 89.57 | 40.30 | 88.37 | 49.82 | 89.97 | 45.76 | 98.42 | 6.61 | 88.35 | 49.70 | 90.93 | 38.44 |
| + Energy | 90.32 | 37.02 | 88.92 | 46.67 | 90.65 | 42.09 | 98.72 | 5.21 | 89.23 | 45.60 | 91.57 | 35.32 |
| + Mahalanobis | 92.06 | 29.38 | 89.41 | 46.03 | 91.53 | 39.21 | 99.20 | 3.07 | 90.76 | 41.65 | 92.59 | 31.87 |
| + Residual | 89.31 | 43.60 | 82.48 | 65.82 | 86.52 | 56.29 | 97.04 | 14.73 | 82.29 | 62.05 | 87.53 | 48.50 |
| + Vim | 90.62 | 35.25 | 89.11 | 45.87 | 90.88 | 41.07 | 98.81 | 4.86 | 89.42 | 44.15 | 91.77 | 34.24 |
| + NECO | 90.47 | 35.04 | 88.46 | 46.25 | 90.73 | 39.93 | 98.82 | 4.56 | 88.81 | 45.15 | 91.46 | 34.19 |
| + X-Maha (ours) | 95.52 | 16.03 | 89.27 | 46.36 | 91.57 | 38.40 | 99.25 | 3.07 | 92.55 | 35.25 | **93.63** | **27.82** |
| **LoRA** | | | | | | | | | | | | |
| + MSP | 85.99 | 47.75 | 85.29 | 56.70 | 86.36 | 53.65 | 97.14 | 11.87 | 83.59 | 58.30 | 87.67 | 45.65 |
| + MLS | 90.06 | 39.08 | 88.56 | 50.13 | 90.17 | 45.98 | 98.41 | 6.79 | 88.82 | 48.35 | 91.20 | 38.07 |
| + Energy | 90.81 | 35.80 | 89.03 | 47.35 | 90.81 | 42.37 | 98.70 | 5.17 | 89.78 | 43.35 | 91.93 | 34.81 |
| + Mahalanobis | 93.12 | 25.78 | 88.31 | 50.24 | 90.92 | 41.66 | 99.28 | 2.84 | 91.57 | 39.00 | 92.64 | 31.90 |
| + Residual | 91.25 | 37.61 | 78.95 | 71.78 | 84.13 | 61.05 | 97.08 | 14.96 | 83.92 | 59.10 | 87.07 | 48.90 |
| + Vim | 91.18 | 33.92 | 89.16 | 46.63 | 91.01 | 41.06 | 98.81 | 4.81 | 90.01 | 42.30 | 92.03 | 33.74 |
| + NECO | 91.80 | 30.76 | 88.35 | 47.11 | 90.79 | 40.31 | 98.93 | 4.07 | 89.71 | 43.80 | 91.92 | 33.21 |
| + X-Maha (ours) | 96.85 | 11.28 | 88.36 | 49.55 | 91.06 | 40.36 | 99.26 | 2.84 | 93.58 | 30.70 | **93.82** | **26.95** |
| **Adapter** | | | | | | | | | | | | |
| + MSP | 85.48 | 49.04 | 84.97 | 56.62 | 86.28 | 53.16 | 96.97 | 12.59 | 83.56 | 57.50 | 87.45 | 45.78 |
| + MLS | 89.75 | 40.18 | 88.51 | 49.51 | 90.28 | 44.91 | 98.34 | 6.89 | 88.88 | 48.10 | 91.15 | 37.92 |
| + Energy | 90.47 | 37.02 | 89.01 | 46.88 | 90.93 | 41.73 | 98.65 | 5.59 | 89.79 | 42.90 | 91.77 | 34.82 |
| + Mahalanobis | 92.61 | 28.32 | 89.17 | 47.15 | 91.47 | 39.20 | 99.24 | 3.00 | 91.35 | 39.95 | 92.77 | 31.52 |
| + Residual | 91.32 | 37.02 | 82.47 | 65.59 | 86.63 | 55.03 | 97.42 | 12.88 | 83.67 | 60.25 | 88.30 | 46.15 |
| + Vim | 90.83 | 35.55 | 89.20 | 45.93 | 91.15 | 40.46 | 98.76 | 5.14 | 90.00 | 42.05 | 91.99 | 33.83 |
| + NECO | 91.13 | 33.10 | 88.91 | 44.55 | 91.23 | 37.50 | 98.84 | 4.33 | 89.49 | 43.10 | 91.92 | 32.52 |
| + X-Maha (ours) | 96.71 | 12.48 | 89.35 | 46.41 | 91.87 | 37.28 | 99.28 | 2.77 | 93.41 | 31.90 | **94.12** | **26.17** |
| **Full fine-tuning** | | | | | | | | | | | | |
| + MSP | 82.21 | 56.24 | 81.12 | 65.98 | 81.83 | 62.82 | 93.92 | 24.60 | 78.67 | 64.85 | 83.55 | 54.90 |
| + MLS | 87.34 | 48.72 | 84.31 | 60.57 | 85.87 | 57.24 | 96.04 | 18.77 | 86.30 | 56.00 | 87.97 | 48.26 |
| + Energy | 87.32 | 49.65 | 84.07 | 61.19 | 85.70 | 58.51 | 95.57 | 22.25 | 86.55 | 55.45 | 87.84 | 49.41 |
| + Mahalanobis | 89.84 | 37.94 | 85.82 | 56.61 | 87.51 | 53.62 | 98.21 | 7.27 | 87.21 | 52.45 | 89.72 | 41.28 |
| + Residual | 81.65 | 65.37 | 71.82 | 83.97 | 75.22 | 78.32 | 92.41 | 38.93 | 70.36 | 79.90 | 78.29 | 69.30 |
| + Vim | 87.56 | 48.42 | 84.12 | 60.91 | 85.80 | 58.11 | 95.81 | 20.61 | 86.53 | 55.70 | 87.97 | 48.75 |
| + NECO | 87.56 | 43.74 | 83.30 | 57.95 | 85.99 | 53.72 | 97.11 | 11.76 | 85.76 | 52.20 | 87.95 | 43.87 |
| + X-Maha (ours) | 93.28 | 25.05 | 86.17 | 55.15 | 87.90 | 51.41 | 98.54 | 5.99 | 88.87 | 46.35 | **90.95** | **36.79** |

