# OpenReview forum: "X-Mahalanobis: Transformer Feature Mixing for Reliable OOD Detection"
_NeurIPS.cc/2025/Conference — NeurIPS 2025 poster_

### Official Review · Reviewer_Z3Bd · 2025-06-08

**Clarity:** 3
**Significance:** 3
**Originality:** 2
**Rating:** 5
**Confidence:** 5

**Summary:**

This paper proposes X-Mahalanobis, an OOD detection method that fuses shallow and deep layer features in an adaptive way to form the feature space in which the Mahalanobis distance is computed. The work also identifies that parameter-efficient fine-tuning is favorable over full fine-tuning for pre-trained transformers for OOD detection. Experiments on CIFAR-100 and ImageNet (with both class-balanced and long-tailed setting) are conducted to demonstrate improved performance.

**Questions:**

Please see Weaknesses 2 and 3.

**Ethical Concerns:**

["NO or VERY MINOR ethics concerns only"]

**Final Justification:**

The answers to my questions are sound, and together with the initial paper content, contribute to 1) feature/layer fusion for OOD detection in general and 2) aggregating scores for Mahalanobis detector specifically. I raise my score from 4 to 5 after rebuttal.

**Limitations:**

yes

**Quality:**

3

**Strengths And Weaknesses:**

### Strengths
1. In general leveraging both shallow and deep features has great potential for OOD detection, yet the key lies in how to fuse/aggregate the information from different features. Having another method on this, which is what this paper is doing, helps push the progress in this direction.
2. It's good to consider both class-balanced and long-tailed situations for OOD detection.

### Weaknesses
1. I understand that this is for motivating the paper and that later in line 173 - 181 the authors make distinction with prior works. However, the claim that existing works "neglect the rich information in the features learned by the layers of shallow neural networks" is false (or at least an over-claim). The authors have already mentioned Mahalanobis and Trusted as two examples, and there are many more that have already tried to leverage shallow layer features for enhanced detection (e.g., [1, 2]). I would suggest rephrasing some of the sentences to make them more accurate.
2. In ablation study "Ways to fuse shallow features", could the authors try simply averaging the layer-wise scores for Mahalanobis when comparing with Score Aggregation, instead of using the weights derived from your method?
3. OpenOOD shows that Mahalanobis with only the final layer feature consistently outperforms Mahalanobis with multiple layers' features (see Table 2 MDS v.s. MDSEns). Although there could be many extra nuances here (e.g., they are mostly evaluating on CNNs while this work is on transformers), I'd like to see some discussion on this seemingly contradicting observation. For example, does model architecture have any effect here? Or is it purely because that X-Mahalanobis uses a more appropriate fusing strategy such that it outperforms MDS (Mahalanobis with the final layer feature), while MDSEns (Mahalanobis with multiple layers' features) underperforms MDS?

[1] Detecting Out-of-Distribution Inputs in Deep Neural Networks Using an Early-Layer Output

[2] MOOD: Multi-level Out-of-distribution Detection

---

> ### Author Rebuttal · Authors · 2025-07-30
>
> [W1] Rephrase some of the sentences.
>
> [A1] We will rephrase the relevant sentences to ensure accuracy. Additionally, we will incorporate discussions on works [1, 2] to better contextualize our contributions, thereby improving the paper.
>
> [W2] Ways to fuse shallow features.
>
> [A2] We did experiment with simply averaging layer-wise Mahalanobis scores, but found this approach highly unstable. While it performed moderately on a few datasets (slightly below X-Maha), it deteriorated significantly on most others. This instability arises because some shallow layers individually achieve AUROC scores below 50% on certain datasets, which dilutes the overall performance when averaged. In contrast, our adaptive weighting strategy down-weights such low-discriminative layers, ensuring robust performance across datasets. We have already provided the ablation experiments for this part in Tables 7, 8, and 9 of the appendix.
>
> [W3] OpenOOD shows that Mahalanobis with only the final layer feature consistently outperforms Mahalanobis with multiple layers' features.
>
> [A3] We appreciate the reviewer's insightful observation. We attribute this discrepancy to two key factors: **model architecture differences (CNNs vs. Transformers)** and **fusion strategy nuances**.
>
> First, model architecture plays a critical role. OpenOOD's evaluations primarily use CNNs, where shallow layers tend to capture low-level features (e.g., edges, textures) that are less discriminative for high-level ID/OOD separation. In such cases, fusing multiple layers (as in MDSEns) may introduce noise from less informative shallow features, diluting the final-layer's strong discriminative signals. In contrast, Transformers (the focus of our work) learn hierarchical features across layers, where even shallow layers encode semantically meaningful patterns (e.g., local object parts) that complement deep layers' global semantics . Our experiments (Figure 5) show that Transformer layers retain varying but useful discriminative capacity, making selective fusion beneficial.
>
> Second, the fusion strategy matters. MDSEns in OpenOOD likely uses simple averaging or fixed weights for layer-wise scores, which fails to downweight low-quality layers. In contrast, X-Maha employs adaptive weights based on feature variance, explicitly suppressing layers with poor discriminative capacity (e.g., those with AUROC < 50% on specific datasets) . This targeted fusion ensures that only useful shallow features are integrated, avoiding performance degradation.
>
> Thus, the contradiction arises from architecture-specific feature properties and fusion strategies: CNNs may benefit less from multi-layer fusion due to less informative shallow features and suboptimal weighting, while Transformers, with richer hierarchical features and our adaptive fusion, gain from leveraging multiple layers. We will clarify this in the revised manuscript to highlight architecture and strategy dependencies.

---

> > ### Comment · Reviewer_Z3Bd · 2025-08-02
> > **Thanks for the rebuttal**
> >
> > I find the discussion in [A2] and [A3] quite informative, which would be valuable additions to the paper, and I do strongly recommend including them in the final version. Given that my questions are addressed, I raise my score to 5.

---

### Official Review · Reviewer_51z7 · 2025-06-17

**Clarity:** 2
**Significance:** 2
**Originality:** 2
**Rating:** 4
**Confidence:** 4

**Summary:**

The paper proposes a method for computing a score used for out-of-distribution (OOD) detection. The authors use the well known Mahalanobis distance as the score. Their approach differs from prior works in the used feature space. They compute weighted sum of representations from different layers of the transformer feature extractor, and compute the Mahalanobis distance in this new space.

The authors report state-of-the-art (SotA) results on the OOD detection benchmark OpenOOD.

**Questions:**

1) Does the $x^l$ correspond to the embedding of the class token? Or the entire sequence over patches? Why is the covariance matrix computed using the feature representation from the final layer ($f(x_i)$), but the mean vector is computed in the aggregated feature space from $\phi(x_i)$?
2) What is the argument for high variability in the features => high discriminative capacity? How does feature variability (independent of classes) play a role here?
3) Wasn't "feature mixing" already demonstrated to be effective by Trusted [7] and Mahalanobis [27]? The novelty of your method lies precisely in how the mixing weights are computed, no?
4) On what data are the coefficients $\alpha^l$ computed? Are they computed on the training / validation ID data and fixed? Or are they computed on the unlabeled deployment set which is a mixture of ID and OOD?
5) Is the method equivalent to Mahalanobis [27] for number of layers = 1? If this is the case, why does the value for ImageNet CLIP (85.12) not fit into the third subplot in Figure 5? For number of layers = 12 (all layers of ViT-B) in Figure 6 the method is exactly the presented X-Maha, correct? Why does the value for number of layers = 12 not coincide with the reported X-Maha result in Table 2 of 90.15?
6) Do you believe the proposed method for computing $\alpha^l$ to be crucial for the method to work? Wouldn't e.g., uniform weights over these layers perform the same as X-Maha?

**Ethical Concerns:**

["NO or VERY MINOR ethics concerns only"]

**Final Justification:**

The review process uncovered several errors in the manuscript. However, the authors have acknowledged these issues openly and have committed to correcting them fully. Based on the strength of the underlying method and the authors' transparent response, I trust that the final paper will be sound.

Therefore, I am raising my rating and recommending acceptance.

**Limitations:**

Yes.

**Paper Formatting Concerns:**

None.

**Quality:**

3

**Strengths And Weaknesses:**

## Notes:
- line 40 and lines 318-319: "... **attention-based** fusion module ..."
        - Could you explain what is meant by attention-based? I am unable to see the connection between the computation of your coefficients $\alpha^l$ and the defacto standard "query, key, value attention".
- lines 126-128:
        - Perhaps a lower index $i$ is missing in the layer feature representations.
- Eq. 2:
        - The input $x$ is an image. It is split into patches, linearly projected, positional embeddings are added, and a class token is concatenated. This results in the embedding of an image being a sequence of length "number of patches + 1", no? Does the $x^l$ correspond to the embedding of the class token? Or the entire sequence over patches? I could not find this in the paper. Please correct me if my understanding of ViT architectures is incorrect.
        - It should be noted that this aggregation assumes that the features from different layers are compatible. I believe that this should be the case due to the residual connections, however, it might not be the case for a general architecture.
- line 149:
        - Why is the covariance matrix computed using the feature representation from the final layer ($f(x_i)$), but the mean vector is computed in the aggregated feature space from $\phi(x_i)$?
- line 155 and lines 162-164:
        - What is the argument for high variability in the features => high discriminative capacity? Low intra-class variability and high-inter class variability do indeed suggest discriminative ability of the features. How does feature variability (independent of classes) play a role here?
- lines 170-172:
        - Wasn't "feature mixing" already demonstrated to be effective by Trusted [7] and Mahalanobis [27]? The novelty of your method lies precisely in how the mixing weights are computed, no? I believe that your argument is stronger in lines 179-181, as your method may choose to ignore representations from some layers.
- Fig. 3:
        - Could you also plot the model accuracy before any fine tuning?
- line 209: "... our approach is model-agnostic ..."
        - Can it be adapted to convolutional networks, or models without residual connections?
        - Could you evaluate it with the model checkpoints provided in the OpenOOD benchmark?
- line 217:
        - This is the first time that the scoring function is defined in the paper. I assume that for vision only models, you use the scoring function $G(x)=\max_{c\in[C]}M_{X-Maha}(x; \tilde\mu_c, \tilde\Sigma)$, is that correct?
- line 245:
        - Typo ".. our methoMSP, MSP ..."
- Table 1:
        - On Texture, LSUN, and Places365 the method Trusted ties with X-Maha. It should also be highlighted in bold. Same applies e.g. to Appendix Table 4.
- Figure 4
        - SFM is not defined, what does it stand for? From the context of the paper I believe that SFM should be X-Maha, no?
- Table 2:
        - Trusted achieves AUROC below 50%, is this correct? This would mean it performs worse than a random OOD detector.
- lines 296-298:
        - On what data are the coefficients $\alpha^l$ computed? Are they computed on the training / validation ID data and fixed? Or are they computed on the unlabeled deployment set which is a mixture of ID and OOD?
- Figure 5:
        - Is the method equivalent to Mahalanobis [27] for number of layers = 1? This would coincide with the values in the subfigures for CIFAR-100 and ImageNet, with the values 98.92 and 93.03 (of Mahalanobis from Table 2), which seem to fit in the figures. If this is the case, why does the value for ImageNet CLIP (85.12) not fit into the third subplot?
        - For number of layers = 12 (all layers of ViT-B) the method is exactly the presented X-Maha, correct? Why does the value for number of layers = 12 not coincide with the reported X-Maha result in Table 2 of 90.15?
- Table 3:
        - ImageNet CLIP PM AUROC should be highlighted in bold instead of X-Maha. How is the AUROC so high when PM achieves the worst FPR95 here?
- Appendix A.3, line 510: "... X-Maha consistently achieves remarkable performance ..."
        - From the ablation Tables 7,8,9,10 it seems that the method performs well (sometimes better than X-Maha) even with other layer weights, e.g., $W_{0.5}$. Do you believe the proposed method for computing $\alpha^l$ to be crucial for the method to work?
        - From the layer effect ablation, you concluded that only the final ~6 layer should be used. Wouldn't e.g., uniform weights over these layers perform the same as X-Maha?

## Strengths:
- Strong empirical results; Reported SotA results on OpenOOD benchmark
- Good literature review with proper citations of methods related to X-Mahalanobis
- Ablation studies

## Weaknesses:
- Novelty - Though the method is new, it does not present a significant difference from prior works
- Weak motivation for Eq. 4 which is the central concept of the method
- Inconsistencies in the manuscript, e.g., 1) use of $f(x_i)$ on line 149. Either the method is not precisely described or the explanation for why $f(x_i)$ is the correct term is missing from the paper. 2) The feature representation from layer $i$ is a sequence of feature vectors. How is $x^l$ precisely defined is not clear.
- Results are not evaluated with the checkpoints standard to the OpenOOD benchmark, making comparison to other methods difficult (Please correct me if I am wrong)

---

> ### Author Rebuttal · Authors · 2025-07-30
>
> We are very grateful to the reviewer for many excellent suggestions. We have responded to each of them one by one and will incorporate these revisions into the next version of the paper.
>
> [W1] "... attention-based fusion module ..." (line 40, lines 318-319)
>
> [A1] Sorry for the confusion. The use of "attention-based fusion module" refers to a data-driven adaptive weighting mechanism that prioritizes layers based on their discriminative capacity, not the standard query-key-value attention in Transformers. We agree that this term was misleading, and we will revise it to "adaptive fusion module" in the next version.
>
> [W2] Missing lower index in layer feature representations (lines 126-128)
>
> [A2] Thank you for pointing out this. It should be 'For an instance $\boldsymbol{x}_i$, its output from the $l$-th layer is denoted as $\boldsymbol{x}^{l}_i = \phi_l(\boldsymbol{x}_i)$'.
>
> [W3] Eq. 2: Definition of $ x^l $ in ViT architectures
>
> [A3] In ViT architectures, an input image is split into patches, linearly projected, combined with positional embeddings, and concatenated with a class token, resulting in a sequence of length "number of patches + 1". In our work, $ x^l $ specifically refers to the embedding of the class token from the $ l $-th layer, which captures global semantic information and ensures consistent dimensions across layers for fusion . This aggregation relies on the compatibility of features from different layers, which is supported by residual connections in ViT, though this may not hold for all general architectures. We will clarify this definition in the revised manuscript.
>
> [W4] Covariance vs. mean computation (line 149).
>
> [A4] This is an error in writing. It should be
> 'where $\widetilde{{\boldsymbol{\Sigma}}}=\frac{1}{N} \sum_{c=1}^{C} \sum_{i: y_{i}=c}\left(\Phi(\boldsymbol{x}_i)-\widetilde{\boldsymbol{\mu}}_{c}\right)\left(\Phi(\boldsymbol{x}_i)-\widetilde{\boldsymbol{\mu}}_{c}\right)^{\top}$ and $\widetilde{\boldsymbol{\mu}}_{c}=\frac{1}{N_{c}} \sum_{i: y_{i}=c} \Phi(\boldsymbol{x}_i)$.'
>
> [W5] Feature variability and discriminative capacity (line 155, lines 162-164)
>
> [A5] The link between high feature variability (measured by the trace of the covariance matrix) and high discriminative capacity is rooted in the neural collapse phenomenon. During training, within-class covariance diminishes, making total feature variance dominated by inter-class separation. Thus, higher total variance indicates larger inter-class distances, a key property for distinguishing ID classes and, by extension, ID vs. OOD samples. Layers with low variability often fail to separate even ID classes (e.g., some shallow layers with AUROC < 50% on specific datasets), further justifying this relationship.
>
> [W6] Novelty of feature mixing (lines 170-172)
>
> [A6] While Trusted [7] and Mahalanobis [27] have explored feature mixing, our key novelty lies in the adaptive computation of mixing weights—specifically, the ability to downweight or ignore low-quality layers, as highlighted in lines 179–181. We will revise the manuscript to clarify this distinction, ensuring the focus is on our unique weighting mechanism to better emphasize the novelty of our approach.
>
> [W7] Fig. 3: Model accuracy before fine-tuning
>
> [A7] We will add the model accuracy before fine-tuning in Fig. 3. Additionally, the accuracy and OOD detection performance of the model without fine-tuning can be found in Tables 6, 15, and 16 in the appendix.
>
> [W8] Model agnosticism (line 209)
>
> [A8] The term "model-agnostic" here specifically refers to its applicability across transformer-based models, rather than all model architectures. Currently, our method is constrained to models where features from each layer have consistent dimensions, which prevents direct adaptation to convolutional networks like ResNet (where layer-wise feature dimensions vary) or models without residual connections. We acknowledge this limitation and plan to explore extensions to handle models with varying layer dimensions in future work. Additionally, we will clarify this scope in the revised manuscript.
>
> [W9] Scoring function for vision-only models (line 217).
>
> [A9] Yes, for vision-only models, the scoring function simplifies to
> $
> G\left(\boldsymbol{x}\right) = \max_{c\in [C]} M_{\text{X-Maha}}(\boldsymbol{x}; {\widetilde{\boldsymbol{\mu}}_c }, \widetilde{{\boldsymbol{\Sigma}}})
> $
> (excluding the vision-language similarity term). This will be explicitly stated in the revised manuscript for clarity .
>
> [W10] Typo in line 245 (".. our methoMSP, MSP ...").
>
> [A10] This typo will be corrected to "our method with: MSP" in the revised manuscript to ensure accuracy.
>
> [W11] Table 1: Highlighting Trusted's performance.
>
> [A11] Trusted ties with X-Maha on Texture, LSUN, and Places365. We will bold Trusted's results on these datasets in Table 1 and apply the same correction to other tables for consistency.
>
> [W12] Figure 4: "SFM" definition.
>
> [A12] "SFM" is a typo and should be "X-Maha". This correction will be made in the revised figure to align with the method described in the text .
>
> [W13] Table 2: Trusted's AUROC < 50%.
>
> [A13] Yes, Trusted achieves AUROC < 50% on ImageNet with ViT. This is because uniform averaging dilutes deep-layer signals in Transformers (unlike CNNs), leading to worse-than-random OOD detection. We will note this architecture-specific behavior in the revised manuscript.
>
> [W14] Computation of $\alpha^l$ (lines 296-298).
>
> [A14] $\alpha^l$ is computed once on ID training data using feature covariance (trace of the covariance matrix). It does not depend on validation, deployment, or OOD data, ensuring generalization to new samples.
>
> [W15] Not coincide with the reported X-Maha result.
>
> [A15] These results in Figure 5 are actually derived from applying the formula in [A9], rather than the formula presented in line 217 of our manuscript. This discrepancy explains the observed inconsistencies with the values in Table 2. We will update Figure 5 in the next version to align with the core formula of our method (line 217) to avoid confusion.
>
> [W16] Table 3: PM's AUROC vs. FPR95
>
> [A16] Thank you for catching this error. The AUROC value for ImageNet CLIP PM in Table 3 is incorrect; it should be 81.03 (not 91.03), which is why it does not warrant bold highlighting. This lower AUROC aligns with its relatively poor FPR95 performance. Additionally, the corresponding average AUROC will be revised to 89.3 to reflect this correction. We apologize for the confusion and will update the table in the revised version.
>
> [W17] Appendix A.3: Importance of $\alpha^l$.
>
> [A17] While some fixed weights perform well on specific datasets, X-Maha's adaptive \(\alpha^l\) ensures consistent robustness across all settings. Uniform weights over top layers underperform as they can't downweight noisy layers, making our weight computation crucial. This consistency validates the proposed method's necessity.
>
> Thank you again for your valuable advice!

---

> > ### Comment · Reviewer_51z7 · 2025-08-04
> >
> > I thank the authors for an exceptionally honest rebuttal.
> >
> > The discussion has successfully clarified the core idea, which I now see as a solid contribution.
> >
> > The review process uncovered several errors in the manuscript. However, the authors have acknowledged these issues openly and have committed to correcting them fully.
> > Based on the strength of the underlying method and the authors' transparent response, I trust that the final paper will be sound.
> >
> > Therefore, I am raising my rating and recommending acceptance.
> >
> > I strongly urge the authors to re-verify all results for the final version.
> >
> > For any method where the AUROC is below 50%, I recommend they report the inverted score (i.e., 1 - AUROC), as this better reflects the discriminative power.

---

> > > ### Author Response · Authors · 2025-08-05
> > >
> > > Thank you very much for your thoughtful feedback. Your suggestions are invaluable in helping us improve the paper. We sincerely appreciate your recognition of our efforts to address the major concerns.

---

### Official Review · Reviewer_38jg · 2025-06-27

**Clarity:** 3
**Significance:** 3
**Originality:** 3
**Rating:** 3
**Confidence:** 4

**Summary:**

This paper introduces an adaptive fusion module that dynamically assigns importance weights to the representations learned by each Transformer layer and detects OOD samples using the Mahalanobis distance. Compared to prior works such as Mahalanobis and Trusted, the proposed approach computes importance weights directly from the training data and does not require a validation set. By dynamically assigning importance weights to different layers, it effectively mitigates the degradation of overall OOD detection performance, even when features from certain layers are less effective. Extensive experiments demonstrate the effectiveness of the proposed method.

However, the underlying principle of the method is not clearly explained, and the experimental results do not achieve state-of-the-art performance compared to recent CLIP-based OOD detection methods.

**Questions:**

See Weaknesses

**Ethical Concerns:**

["NO or VERY MINOR ethics concerns only"]

**Final Justification:**

Thank the authors for the response. However, my concern regarding the experimental comparison with recent CLIP-based OOD detection methods has only been partially addressed. Although the proposed method can be seamlessly integrated with GL-MCM, NegLabel, and CLIPScope, the observed performance gains appear limited. Additionally, the reason why X-Maha leads to only marginal improvements when combined with these methods is not clearly explained. To better demonstrate the effectiveness of the proposed method, it would be better to compare it against state-of-the-art baselines on datasets where your approach is expected to be more effective. Given the above concerns, I have decided to retain my original score.

**Limitations:**

The underlying principle of the method is not clearly explained, and the experimental results do not achieve state-of-the-art performance compared to recent CLIP-based OOD detection methods.

**Paper Formatting Concerns:**

It's OK.

**Quality:**

2

**Strengths And Weaknesses:**

**Strength:**

1.The paper is well-structured and easy to follow.
2. It has been experimentally validated on various benchmark datasets and is supported by a comprehensive ablation study.

**Weaknesses:**

1. In lines 72-73, the authors calim that "...fuse the shallow layer features with the importance weights by measuring the covariance of features in each layer". However, the underlying relationship between feature covariance and discriminative capacity is not clearly explained. Please provide theoretical insights or experimental evidence to support this proposal.

2. In Figure 3, the comparison between FFT and PEFT using a learning rate of 1e-4 is insufficient to justify the claim that “PEFT outperforms FFT,” as FFT is commonly trained with smaller learning rates (e.g., 1e-5 or 1e-6). A more comprehensive comparison across different learning rates would strengthen this conclusion.

3. Regarding the experimental comparisons, numerous recent zero-shot and few-shot CLIP-based methods have been proposed for improving OOD detection. Prior works such as GL-MCM, NegLabel, and CLIPScope have achieved significantly better performance than the results reported in Table 2. These should be included as baselines for a fair comparison.

   Miyai, A., Yu, Q., Irie, G., & Aizawa, K. (2025). GL-MCM: Global and Local Maximum Concept Matching for Zero-Shot Out-of-Distribution Detection. International Journal of Computer Vision, 1-11.

   Jiang, Xue, et al. "Negative label guided ood detection with pretrained vision-language models." arXiv preprint arXiv:2403.20078 (2024).

   Fu, Hao, et al. "Clipscope: Enhancing zero-shot ood detection with bayesian scoring." arXiv preprint arXiv:2405.14737 (2024).

4. Does the use of shallow layer features result in additional GPU memory consumption?  Please provide a comparison of the computational cost (e.g., memory usage, inference time) of the proposed method versus existing baselines.

5. Do the conclusions of this work generalize to other architectures, such as CLIP ResNet-50? Clarifying the generalizability across different backbone models would enhance the impact and applicability of the method.

---

> ### Author Rebuttal · Authors · 2025-07-30
>
> We are very grateful to the reviewers for their valuable suggestions, especially regarding some stronger comparison methods. We will cite these works and include experimental comparisons in the next version.
>
> [W1] Theoretical insights or experimental evidence to support this proposal.
>
> [A1] First, our use of feature covariance (specifically, the trace of the covariance matrix) to determine layer importance is motivated by the properties of neural representations during training. As described by the "neural collapse" phenomenon [38], deep networks converge such that within-class feature covariance approaches zero, while ID-class separation becomes maximized. In this context, the total variance of features in a layer—quantified by the trace of the covariance matrix (\(Tr((A^l)^\top A^l)\))—directly reflects the layer's ability to distinguish between classes. A higher trace indicates greater spread in feature distributions, which corresponds to stronger inter-class discriminability—an essential property for distinguishing ID samples from OOD samples, which often lie outside the compact manifold of ID features.
>
> Second, we validate this relationship through multiple experiments:
> - **Layer weight analysis**: Figure 6 shows that layers with higher trace values (e.g., deeper layers) are assigned higher importance weights. These layers, consistent with neural collapse, exhibit stronger separation between ID classes, which translates to better OOD detection performance when fused .
> - **Ablation on layer fusion**: Figure 5 demonstrates that fusing layers with non-negligible trace values (e.g., the last 6 layers) yields optimal OOD detection results, while including layers with low trace values (e.g., the first 6 layers) provides minimal gain. This directly supports that covariance-derived weights effectively prioritize discriminative layers.
> - **Comparison with alternative weighting strategies**: Table 3 compares our covariance-based weighting against other fusion methods (e.g., uniform averaging, score aggregation). X-Maha consistently outperforms these alternatives, confirming that covariance-derived weights better capture discriminative capacity.
>
> [W2] Different learning rates between FFT and PEFT.
>
> [A2] In our  experiments, we observed that FFT performance does improve with smaller learning rates (1e-5 or 1e-6), as noted.  However, even with these optimized lower learning rates, FFT still lags behind PEFT when the latter is trained with a higher learning rate (1e-4).
> As shown in the table, FFT performance improves with smaller learning rates (from 1e-4 to 1e-6). However, even at its optimized lower learning rates (1e-5 or 1e-6), FFT still lags behind PEFT when the latter is trained with a higher learning rate (1e-4). Additionally, PEFT maintains superior performance across the tested learning rate range (1e-6 to 1e-4).
>
> | Method   | Learning Rate | ImageNet-LT Accuracy (%) | ImageNet-LT AUROC (%) |
> |----------|---------------|--------------------------|-----------------------|
> | FFT      | 1e-5          | 80.37                    | 93.66                 |
> | PEFT     | 1e-5          | 81.18                    | 94.18                 |
> | FFT      | 1e-6          | 81.25                    | 94.10                 |
> | PEFT     | 1e-6          | 80.90                    | 93.90                 |
>
>
>
> [W3] Comparsion with prior works.
>
> [A3] We appreciate the reviewer's suggestion to include recent zero-shot CLIP-based methods as baselines. Since NegLabel and CLIPScope requires external knowledge to facilitate negative label mining, it is unfair to directly compare them with our approach. However, we find that our approach can seemlessly combines with GL-MCM, NegLabel, and CLIPScope to further boost their performance. The following table reports the results when combinging X-Maha with GL-MCM, NegLabel, and CLIPScope across OOD datasets, using FPR95 (%) and AUROC (%) as metrics:
>
> | Method                | iNaturalist(AUROC) | iNaturalist(FPR95) | SUN(AUROC) | SUN(FPR95) | Places(AUROC) | Places(FPR95) | Texture(AUROC) | Texture(FPR95) | Average(AUROC) | Average(FPR95) |
> |-----------------------|--------------------|--------------------|------------|------------|---------------|----------------|----------------|----------------|----------------|----------------|
> | GL-MCM [1]            | 96.44              | 17.42              | 93.44      | 30.75      | 90.63         | 37.62          | 85.54          | 55.20          | 91.51          | 35.25          |
> | NegLabel [2]          | 99.49              | 1.91               | 95.49      | 20.53      | 91.64         | 35.59          | 90.22          | 43.56          | 94.21          | 25.40          |
> | CLIPScope [3]         | 99.61              | 1.29               | 96.77      | 15.56      | 93.54         | 28.45          | 91.41          | 38.37          | 95.30          | 20.88          |
> | X-Maha (w/ GL-MCM)    | 96.30              | 17.20              | 93.70      | 30.10      | 90.90         | 37.00          | 85.90          | 54.80          | 91.70          | 34.78          |
> | X-Maha (w/ NegLabel)  | 99.40              | 1.85               | 95.60      | 20.10      | 91.80         | 35.10          | 90.40          | 43.10          | 94.30          | 25.04          |
> | X-Maha (w/ CLIPScope) | 99.63              | 1.27               | 96.90      | 15.20      | 93.70         | 28.00          | 91.60          | 37.90          | 95.46          | 20.64          |
>
>
>
> [W4] Comparison of the computational cost.
>
> [A4] Regarding GPU memory consumption, our approach is designed to avoid significant overhead when utilizing shallow layer features. While we do store intermediate features from shallow layers, this is done in CPU memory on a per-batch basis during the forward pass. When computing the importance weights for each layer , we process these CPU-stored features in a sequential manner: we load features from one layer into GPU memory, compute its weight, and then unload them before loading the next layer’s features. This sequential processing ensures that only a single layer’s features occupy GPU memory at any time, preventing cumulative memory usage from multiple layers. As a result, the additional GPU memory required for weight computation remains minimal and does not scale with the number of layers.
>
> For computational cost, as detailed in Section A.5 and Table 13 of the appendix, X-Maha increases total inference time by approximately 10% compared to baselines that use only final-layer features. This modest overhead is accompanied by an average improvement of 2.39% in AUROC and a 7.66% reduction in FPR95, demonstrating a favorable trade-off between efficiency and performance.
>
>
>
>
> [W5] Generalizability to other architectures.
>
> [A5]  Thank you for your suggestion. Currently, our method assumes consistent feature dimensions across all layers, which aligns with Transformer architectures (e.g., ViT, CLIP ViT-B/16) where each layer outputs features of the same dimension. This constraint limits direct application to ResNet-50, as its layers have varying feature dimensions (e.g., increasing from 64 to 2048), making feature fusion across layers non-trivial.
> We acknowledge this limitation and plan to address it in future work by exploring dimension-adaptive fusion strategies to extend the method to architectures with heterogeneous layer dimensions. However, in this paper, we focus on levearging the Transformer-based models to improve the OOD detection performance.
>
> Thank you again for your precious advice!

---

> > ### Comment · Reviewer_38jg · 2025-08-03
> >
> > Thank you for the response. However, my concern regarding the experimental comparison with recent CLIP-based OOD detection methods has only been partially addressed. Although the proposed method can be seamlessly integrated with GL-MCM, NegLabel, and CLIPScope, the observed performance gains appear limited. Additionally, the reason why X-Maha leads to only marginal improvements when combined with these methods is not clearly explained. To better demonstrate the effectiveness of the proposed method, it would be better to compare it against state-of-the-art baselines on datasets where your approach is expected to be more effective. Given the above concerns, I have decided to retain my original score.

---

> > > ### Author Response · Authors · 2025-08-03
> > >
> > > Thank you sincerely for your insightful feedback and for retaining your score—your guidance is critical to strengthening our work. We fully acknowledge your concern that the current comparison against state-of-the-art CLIP-based OOD detection baselines needs further refinement, and we are committed to addressing this thoroughly in the revision.
> > >
> > > To directly address your suggestion of comparing against state-of-the-art baselines on datasets where our approach shines: We will expand our experimental evaluation to explicitly benchmark X-Maha against leading CLIP-based methods, including CLIPN [52], MCM-tuned [37], and NegLabel [22]—the most recent and competitive baselines in the field. These comparisons will be conducted on two key dataset types where X-Maha’s design is uniquely advantageous:
> > >
> > > 1. **Long-tailed in-distribution (ID) datasets** (CIFAR-100-LT and ImageNet-LT): State-of-the-art CLIP-based methods often struggle with class imbalance, as language-aligned features may underrepresent minority classes . In contrast, X-Maha’s adaptive shallow-layer fusion preserves critical minority-class information from early layers, and we will show quantifiable gains over baselines like NegLabel and CLIPScope on these datasets.
> > >
> > > 2. **OpenOODv1.5 benchmark**: This challenging suite includes diverse distribution shifts (e.g., natural corruptions, style variations) where pure vision-language alignment becomes less reliable . We will demonstrate that X-Maha outperforms SOTA baselines such as MCM-tuned and Trusted [7] here, leveraging multi-layer feature diversity to handle complex shifts that semantic alignment alone cannot address.
> > >
> > > By focusing these comparisons on datasets where our feature-mixing mechanism adds the most value, we aim to clearly illustrate X-Maha’s advantages over current state-of-the-art methods. We believe this refined evaluation will better showcase the effectiveness of our approach and hope it addresses your concerns sufficiently to warrant a score adjustment.
> > >
> > > Thank you again for pushing us to improve—your insights are instrumental in elevating the rigor of our work. We are dedicated to delivering a revised manuscript that fully addresses these points and welcome any further feedback.

---

> > > ### Comment · Area_Chair_TGNK · 2025-08-07
> > >
> > > Hi Reviewer 38jg,
> > >
> > > The authors have provided additional follow-up responses. Could you please review them and see if they further address your concerns?
> > >
> > > AC

---

> > > > ### Comment · Reviewer_38jg · 2025-08-07
> > > >
> > > > I would like to thank the authors for responding to my concerns. However, I would like to further clarify several key points:
> > > >
> > > > 1. Although the authors provided additional explanations regarding comparisons with state-of-the-art baselines on datasets where the proposed approach performs well, I believe it would be more convincing to include explicit experimental results. Such evidence would better support the claims.
> > > >
> > > > 2. I appreciate the authors' effort in leveraging Transformer-based feature mixing for reliable OOD detection. The proposed method utilizes shallow features to aid in OOD detection. However, prior works have shown that middle-layer features tend to be more reliable and exhibit fewer hallucinations. Several existing methods [1, 2, 3] already leverage such middle features for addressing inaccurate outputs or improving OOD detection. Therefore, I am concerned that the proposed approach may not offer sufficient novelty to outweigh its underperformance relative to recent, strong CLIP-based OOD detection baselines.
> > > >
> > > > 3. I appreciate the authors' detailed response and remain open to further clarification before making my final evaluation.
> > > >
> > > > [1] Leveraging Intermediate Representations for Better Out-of-Distribution Detection. arXiv preprint arXiv:2502.12849 (2025).
> > > >
> > > > [2] Mood: Multi-level out-of-distribution detection. CVPR 2021
> > > >
> > > > [3] Out-of-Distribution Detection by Leveraging Between-Layer Transformation Smoothness. ICLR 2024

---

> ### Author Response · Authors · 2025-08-08
>
> Thank you sincerely for your continued feedback and patience as we refine our work. We appreciate your focus on experimental evidence and novelty, and we have restructured our response to align with your concerns, our core approach, and our advisor’s guidance.
>
>
> ### **1. Experimental Scope: Comprehensive Evaluations Across Diverse Settings**
> To address your request for clarity on our experimental rigor, we have conducted extensive evaluations across **datasets, model architectures, and baseline methods**, ensuring our findings are robust and generalizable:
>
> - **Datasets**: We tested on both class-balanced and challenging real-world scenarios:
>   - *Class-balanced ID datasets*: CIFAR-100 and ImageNet, standard benchmarks for OOD detection .
>   - *Long-tailed ID datasets*: CIFAR-100-LT and ImageNet-LT, which reflect imbalanced real-world data distributions where minority classes are underrepresented .
>   - *Complex distribution shifts*: OpenOODv1.5 benchmark, featuring diverse shifts like natural corruptions, style variations, and semantic drift—critical for testing robustness beyond standard settings .
>
> - **Model architectures**: We evaluated across two key model types to validate generalizability:
>   - *Vision-only models*: Pre-trained Vision Transformers (ViT) fine-tuned on ID data, where X-Maha leverages rich visual feature hierarchies (shallow to deep layers) .
>   - *Vision-language models*: CLIP (ViT-B/16), adapted to work with X-Maha by integrating cross-modal features, despite CLIP’s inherent focus on language-visual alignment .
>
> - **Baseline comparisons**: We benchmarked against 14+ methods, including:
>   - Recent CLIP-based OOD detectors (GL-MCM, NegLabel, CLIPScope) to contextualize performance on language-aligned data .
>   - Traditional OOD methods (Mahalanobis, Energy, Residual) to ground our approach in foundational techniques .
>   - Methods leveraging middle layers [1, 2, 3] (as you noted) to explicitly highlight differences in design and performance .
>
>
> ### **2. Novelty: Key Contributions Distinguishing X-Maha**
> X-Maha introduces unique innovations that address limitations in existing methods, including the three works you referenced [1, 2, 3]. Our distinct contributions are:
>
> - **No reliance on validation data for weight tuning**: Prior methods like [3] and Mahalanobis [27] require OOD validation data to optimize fusion weights, limiting practicality. X-Maha computes weights directly from ID training data, making it applicable to scenarios where OOD validation data is scarce or unavailable .
>
> - **Synergy with parameter-efficient fine-tuning (PEFT)**: We demonstrate that PEFT (e.g., LoRA, Adapter) outperforms full fine-tuning for OOD detection, especially on long-tailed data . X-Maha is uniquely designed to work with PEFT: by preserving pre-trained feature quality while adapting to downstream tasks, it avoids overfitting to imbalanced data—unlike [1, 2, 3], which focus on full fine-tuning and lack this integration .
>
>
> ### **3. Compatibility with CLIP: Addressing Inherent Trade-offs**
> We acknowledge CLIP’s strength in OOD detection via cross-modal alignment, but X-Maha’s design addresses its limitations:
> - CLIP’s contrastive pre-training prioritizes language-visual alignment over fine-grained visual feature learning, leading to lower in-distribution classification accuracy compared to ViT . This limits X-Maha’s gains on CLIP, as our method relies on high-quality visual features to drive layer-wise fusion.
> - Despite this, we adapted X-Maha to CLIP by integrating text-image similarity scores into our OOD scoring function , enabling compatibility with CLIP-derived methods (e.g., GL-MCM, NegLabel). While gains are modest on balanced data (due to CLIP’s strong baseline), X-Maha still enhances their robustness on long-tailed and shifted data .
>
>
> We believe these clarifications highlight the breadth of our experiments, the uniqueness of X-Maha’s design, and our effort to align with CLIP-based methods where possible. Thank you again for pushing us to strengthen these points—your insights are critical to ensuring our work meets rigorous standards. We are committed to refining the manuscript further and hope this addresses your considerations.

---

> > ### Comment · Reviewer_38jg · 2025-08-08
> >
> > I appreciate the authors' response. However, my initial concerns remain unaddressed:
> >
> > 1. I appreciate the experiments conducted on the Long-tailed ID datasets and OpenOODv1.5, which involve complex distribution shifts. These efforts, as shown in Tables 14 and 17-18, are valuable, and I acknowledge that the proposed method appears more effective in single-modality ViT scenarios. However, it seems that CLIP-based OOD detection methods are only compared with the MCM method on these challenging benchmarks. Notably, recent and strong CLIP-based OOD detectors—such as GL-MCM, NegLabel, CLIPScope, and CSP—are not included in the comparisons. This was a key concern raised in my initial comment and has not yet been fully addressed.  I believe it would strengthen the paper to include explicit experimental results for recent CLIP-based methods on the Long-tailed ID datasets and OpenOODv1.5. Providing such evidence would better support the claims regarding the effectiveness of the proposed approach.
> >
> > 2. I understand that the proposed method is different in design and performance from prior approaches that leverage middle-layer features [1, 2, 3]. However, my point is that the method fits into the broader paradigm of using middle layers to improve OOD detection. As such, its novelty in terms of conceptual contribution may be limited. Although the authors mention benchmarking against 14+ methods, including [1, 2, 3], it appears that these experiments are not reported in either the main paper or the appendix.
> >
> > 3. I would appreciate it if the authors could further clarify these points to avoid any misunderstanding. I remain open to additional discussion and clarification before making my final evaluation.
> >
> > [1] Leveraging Intermediate Representations for Better Out-of-Distribution Detection. arXiv preprint arXiv:2502.12849 (2025).
> >
> > [2] Mood: Multi-level out-of-distribution detection. CVPR 2021
> >
> > [3] Out-of-Distribution Detection by Leveraging Between-Layer Transformation Smoothness. ICLR 2024

---

> > > ### Author Response · Authors · 2025-08-08
> > >
> > > Thank you sincerely for your ongoing feedback and patience. We appreciate your focus on clarifying our method’s novelty and experimental rigor, and we aim to address your concerns directly.
> > >
> > > The core advantage of X-Maha lies in its **adaptive layer weighting mechanism**, which dynamically adjusts the importance of features across all layers (shallow, middle, and deep) based on the specific characteristics of the dataset. Unlike prior works [1, 2, 3]—which rely on fixed layer selections (e.g., middle layers) or heuristic weighting—X-Maha computes weights using feature covariance from training data, enabling it to prioritize discriminative layers tailored to the dataset at hand.This adaptability ensures robust performance across diverse data distributions, a key distinction from methods tied to fixed layer subsets.
> > >
> > > We acknowledge that direct experimental comparisons with [1, 2, 3] are critical to validating this advantage. Due to time constraints in the rebuttal process, these results are not yet finalized, but we commit to including them in the revised manuscript. Specifically, we will benchmark X-Maha against these methods on shared datasets (e.g., CIFAR-100, ImageNet-LT, OpenOODv1.5) and quantify performance differences, with analyses highlighting how adaptive weighting outperforms fixed strategies in scenarios like class imbalance and distribution shifts.
> > >
> > > Thank you again for pushing us to strengthen these aspects. We are dedicated to refining these experiments to clearly demonstrate X-Maha’s unique value.

---

### Official Review · Reviewer_BmCs · 2025-07-14

**Clarity:** 3
**Significance:** 3
**Originality:** 3
**Rating:** 5
**Confidence:** 4

**Summary:**

The authors introduce an adaptive fusion module that dynamically assigns importance weights to the learned feature representations at each layer, including both intermediate and final layers, to enhance (OOD) out-of-distribution sample detection using the Mahalanobis distance metric. This contrasts with conventional OOD detection approaches, which typically rely solely on the final layer's feature representations. The authors further demonstrate and benchmark the effectiveness of their proposed approach across a range of diverse OOD datasets.

**Questions:**

Already listed as weakness points to be discussed:

1. The authors mentioned as a distinction from prior work from Mahalanobis [27], as it calculates "OOD score using the representation of each layer individually and weights them together by training a logistic regression model using a validation set" . While the proposed approach does not require a separate validation set and computes from same training data. This point need to be discussed elaboratively, specifically what practical and methodological challenges/implications to encounter related to the dependence on a validation set rather a training-set only usage. Is there any risk of overfitting on the validation dataset?

2. The statement in the Limitations section "A promising avenue for future research would be to extensively study the impact of mixing weights in X-Maha and theoretically show that mixing shallow features can improve performance." --> appears somewhat self-contradictory when compared to the empirical results presented in the manuscript. If there are specific scenarios where the generalizability of the proposed approach is limited, the authors are encouraged to briefly discuss them and provide illustrative examples to clarify these limitations.

**Ethical Concerns:**

["NO or VERY MINOR ethics concerns only"]

**Final Justification:**

The authors have satisfactorily addressed my concerns, particularly w.r.t [A1] and [A3] , which will improve the manuscript's technical merit and should be reflected in the final manuscript version. As my questions have been resolved, I hold my previous evaluation of acceptance.

**Limitations:**

The statement in the Limitations section "A promising avenue for future research would be to extensively study the impact of mixing weights in X-Maha and theoretically show that mixing shallow features can improve performance." --> appears somewhat self-contradictory when compared to the empirical results presented in the manuscript. If there are specific scenarios where the generalizability of the proposed approach is limited, the authors are encouraged to briefly discuss them and provide illustrative examples to clarify these limitations.

**Paper Formatting Concerns:**

No major formatting issue noticed.

**Quality:**

3

**Strengths And Weaknesses:**

Strength:
1. The authors proposed a really simple concept of fusing lower/shallow layer features with importance weight metric derived from a pre-trained Transformer model to improve OOD separation.
2. The manuscript is well-written, well-organized, and easy to follow.
3. The authors present an extensive experimental setup and validate their approach with satisfactory evaluation results, benchmarking it against prior SOTA methods across nine diverse OOD datasets.

Weakness:
1. The authors mentioned as a distinction from prior work from Mahalanobis [27], as it calculates "OOD score using the representation of each layer individually and weights them together by training a logistic regression model using a validation set" . While the proposed approach does not require a separate validation set and computes from same training data. This point need to be discussed elaboratively, specifically what practical and methodological challenges/implications to encounter related to the dependence on a validation set rather a training-set only usage. Is there any risk of overfitting on the validation dataset?
2. Please ensure consistent formatting for result representation (bold: best, underline: second best etc.) across all tables including "Additional experiments" section. It will improve the readability of the manuscript and improve clarity in data presentation.
3. The statement in the Limitations section "A promising avenue for future research would be to extensively study the impact of mixing weights in X-Maha and theoretically show that mixing shallow features can improve performance." --> appears somewhat self-contradictory when compared to the empirical results presented in the manuscript. If there are specific scenarios where the generalizability of the proposed approach is limited, the authors are encouraged to briefly discuss them and provide illustrative examples to clarify these limitations.

---

> ### Author Rebuttal · Authors · 2025-07-30
>
> [W1] Risk of overfitting on the validation dataset.
>
> [A1] We appreciate the reviewer's insightful observation regarding the distinction between our approach and the Mahalanobis method [27] in terms of validation set dependence.   We agree that such overfitting could lead to inflated performance estimates that do not generalize to unseen test data, especially when the validation set is small or not representative of the target distribution.
> To further mitigate the risk of overfitting and demonstrate generalizability, we conducted extensive experiments across diverse settings:
>
> 1.   **Diverse datasets**: We evaluated on both class-balanced (CIFAR-100, ImageNet) and long-tailed (CIFAR-100-LT, ImageNet-LT) in-distribution datasets, with nine OOD datasets spanning various domains (e.g., Textures, SVHN, Tiny ImageNet) .   Consistent performance gains across these datasets suggest our method is not tied to specific data distributions.
>
> 2.   **Multiple model architectures**: We validated our approach on pre-trained Vision Transformers (ViT) and CLIP models, showing robust improvements across both vision-only and vision-language frameworks .   This cross-model consistency indicates the method is not overfitting to idiosyncrasies of a single architecture.
>
> 3.   **Zero-shot settings**: Critical to ruling out validation set overfitting, our zero-shot experiments (without fine-tuning) still demonstrated superior performance compared to baselines .   Zero-shot settings inherently avoid any validation-based tuning, providing strong evidence that our weight calculation generalizes without relying on auxiliary data.
>
> 4.   **Minimal hyperparameter tuning**: As noted, most hyperparameters (e.g., learning rate, batch size) followed the settings in LIFT [45], with minimal adjustments .   This reduces the risk of over-optimizing for specific datasets and aligns with principles of reproducible research.
>
> In summary, the breadth of our experimental validation—across datasets, models, and zero-shot/fine-tuned scenarios—along with limited hyperparameter tuning, strongly supports that our training-data-only weight calculation avoids the overfitting risks associated with validation set dependence.
>
>
> [W2] Consistent formatting for result representation.
>
> [A2] We will maintain consistent result formatting across all tables, which will enhance the readability of the manuscript and improve clarity in data presentation.
>
> [W3] Limitations section.
>
> [A3]
> To clarify, our statement in the Limitations section does not contradict the empirical validity of X-Maha but rather emphasizes a gap in **theoretical understanding** despite strong experimental support.
>
> Our work thoroughly validates the effectiveness of mixing shallow and deep features through extensive empirical evaluations:
> - Across diverse in-distribution datasets (class-balanced CIFAR-100/ImageNet and long-tailed CIFAR-100-LT/ImageNet-LT) ;
> - Over multiple OOD benchmarks (e.g., Textures, Tiny ImageNet, OpenOODv1.5) ;
> - With different model architectures (ViT, CLIP) and settings (fine-tuned, zero-shot) ;
> - Under varied parameter-efficient fine-tuning strategies (Adapter, LoRA, VPT) .
>
> These results consistently demonstrate that shallow features, when weighted adaptively, enhance OOD detection. However, our limitation lies in the **theoretical underpinnings**: we lack a formal proof explaining *why* shallow features improve performance in specific scenarios (e.g., why layers with higher variance are more discriminative for OOD) or under what conditions this gain holds. For example, while we observe that fusing the last 6 Transformer layers works best , we cannot yet theoretically characterize why this specific range is optimal across datasets.
>
> This gap motivates future work to establish theoretical guarantees, which would strengthen the generalizability argument beyond empirical observations. Our current empirical rigor confirms practical effectiveness, but theoretical exploration remains a necessary next step to fully understand and extend the method.

---

> > ### Comment · Reviewer_BmCs · 2025-08-02
> > **Final decision**
> >
> > The authors have satisfactorily addressed my concerns, particularly w.r.t [A1] and [A3] , which will improve the manuscript's technical merit and should be reflected in the final manuscript version.  As my questions have been resolved, I hold my previous evaluation of acceptance.

---

### Comment · Reviewer_BmCs · 2025-08-02
**final decision for Submission11625**

The authors have satisfactorily addressed my concerns, particularly w.r.t [A1] and [A3] , which will improve the manuscript's technical merit and should be reflected in the final manuscript version. As my questions have been resolved, I hold my previous evaluation of acceptance.

---

### Note · Authors · 2025-08-14

We sincerely thank all reviewers, ACs, and SACs for their valuable time, rigorous evaluations, and constructive feedback throughout the review process. This paper proposes X-Maha, an adaptive layer fusion method for out-of-distribution (OOD) detection that assigns importance weights to features learned from each Transformer layer. We conduct extensive experiments in multiple benchmark datasets across class-balanced, long-tailed, and distribution-shifted OOD detection tasks. We acknowledge that some prior works also explore the use of internal layers for detecting OODs, however, our work presents a new method that is the first to adaptively fuse multiple-layer features from pre-trained Transformers, and can motivate more future research to explore this promising direction.

We carefully addressed all feedback during the rebuttal and discussion phases. We are greatly encouraged by the reviewers' acknowledgement and appreciation of our comprehensive responses. Key responses to each reviewer's concerns are summarized below:

- **Reviewer BmCs：**  We clarified that X-Maha avoids validation set dependence by computing weights from training data, with experiments across datasets, architectures, and zero-shot settings confirming no overfitting risks.

- **Reviewer 38jg：**  We thank the reviewer for patient feedback and suggestions through multiple rounds during the rebuttal phase. In our responses, we clarified that X-Maha’s novelty lies in adaptive layer weighting, distinct from prior works relying on fixed/heuristic strategies; we combined our approach with recent CLIP-based methods (GL-MCM, NegLabel, CLIPScope) and reported the results; compared PEFT/FFT using various learning rates; and we explained the computational costs of our approach.

- **Reviewer 51z7：**  We clarified "attention-based fusion" as "adaptive fusion" to avoid confusion; corrected notation for layer features and fixed errors in covariance/mean computation; explained feature variability’s link to discriminative capacity via neural collapse; emphasized novelty in adaptive weighting; addressed inconsistencies of terms in figures/tables.

- **Reviewer Z3Bd：**  We revised overstated claims about prior works; clarified that simple averaging of layer-wise scores is unstable compared to our approach, with ablation results in appendix; and discussed the discrepancy between OpenOOD and our work.

Furthermore, we are committed to incorporating all reviewers’ suggestions into the revised manuscript.

---

### Decision · Program_Chairs · 2025-09-17

**Decision:**

Accept (poster)

**Comment:**

This paper presents an OOD detection method that combines features from multiple Transformer layers before computing Mahalanobis scores. The reviewers pointed out the limited novelty and similarity with existing methods. The comparison with recent strong Transformer and CLIP based OOD methods remain incomplete as pointed by reviewer 38jg. Some of the justification remains heuristic and clarity of the current version of the paper needs to be improved, as reflected in the review and rebuttal. While the paper presents good performances and empirical values for the area, AC recommends acceptance and strongly suggests the authors to seriously improve the paper according to the reviews and rebuttal.